# TimesMR: Unlocking the Potential of MLP and RNNs for Multivariate Time Series Forecasting

## Abstract

Multivariate time series forecasting is critical across various domains, requiring effective modeling of temporal dependencies and variable correlations. Existing multi-scale models, while effective for temporal dependency modeling, often rely on complex and computationally expensive feature extractors. Similarly, attention mechanisms, though powerful for capturing variable relationships, suffer from high complexity on high-dimensional datasets and introduce noise from weakly related variables, leading to performance degradation. To address these challenges, we propose TimesMR, a novel model with two key innovations. First, we design multi-scale MLP modules, namely multi-patch MLP and multi-downsampling MLP, to enhance temporal dependency modeling with lightweight and efficient architectures. Second, we introduce grouped bidirectional RNNs for efficient variable correlation modeling, which reduce computational costs while preserving performance by grouping variables and capturing both intra- and inter-group correlations. Extensive experiments on sixteen datasets demonstrate that TimesMR achieves state-of-the-art performance, surpassing eighteen existing models. Our contributions include novel plug-and-play modules for temporal and variable modeling, offering an effective and efficient solution for multivariate time series forecasting. The code is available at `https://anonymous.4open.science/r/TimesMR`.

## 1 Introduction

Multivariate time series forecasting is crucial in diverse domains, including weather prediction (Wu et al., 2023b; Bi et al., 2023), traffic management (Yin et al., 2022), market analysis (Liu et al., 2024), disease monitoring (Matsubara et al., 2014), and energy systems (Qian et al., 2019). Effective modeling requires effectively capturing temporal dependencies within sequences and leveraging variable relationship to enhance target variable representation. A unified paradigm has emerged, focusing on two key aspects: 1) Temporal Dependency Modeling: Capturing intricate temporal patterns; 2) Variable Correlation Modeling: Exploring variable interdependencies.

In this work, we address these two aspects by proposing a simple yet effective model, TimesMR, which achieves superior performance. TimesMR integrates a multi-scale MLP module, with two techniques—*multi-patch MLP* and *multi-downsampling MLP*—for temporal dependency modeling, and a variable correlation module utilizing the proposed *grouped bidirectional RNNs* for efficient variable correlation modeling.

**Temporal Dependency Modeling.** Recent studies (Chen et al., 2024b; Wang et al., 2024; Wu et al., 2023a; Wang et al., 2023) highlight the importance of modeling multi-scale features in time series forecasting, as time series data exhibit diverse patterns across different time scales (e.g., minutes, hours, days, months). Multi-scale models enhance feature representation and are primarily implemented in two ways: 1) multi-downsampling, which downsamples the original time series to capture information at varying granularities (e.g., TimeMixer (Wang et al., 2024)); and 2) multi-patch, which introduces multiple patches to integrate multi-scale temporal resolutions and capture diverse patterns (e.g., Pathformer (Chen et al., 2024b)). Interestingly, models like DLinear (Zeng et al., 2023) achieve competitive performance without explicitly using multi-scale techniques, relying instead on Linear transformation applied across all time points. To evaluate the potential of MLPs, we

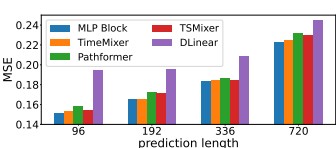

fixed lookback length.

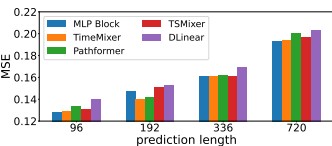

searched lookback length.

Figure 1: MSE evaluation on ECL dataset.

Table 1: Correlation analysis of different datasets.

| Datasets | ETTm1 | ETTm2 | Weather | ECL | Traffic |
|---|---|---|---|---|---|
| MCC | 0.096 | 0.139 | 0.140 | 0.243 | 0.281 |
| Number of variables | 7 | 7 | 21 | 321 | 862 |

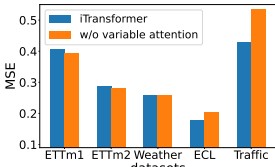 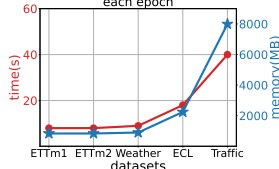

Figure 2: MSE and training cost of iTransformer.

introduce a generic MLP module, termed the MLP block, consisting of a single normalization layer and two fully connected layers. Figure 1 compares the performance of the MLP block with state-of-the-art multi-scale models (TimeMixer, Pathformer, and TSMixer (Ekambaram et al., 2023)) and DLinear on the ECL dataset, using MSE (Mean Squared Error) as the evaluation metric for both fixed and searched lookback lengths. The results demonstrate that the MLP block achieves competitive performance, albeit slightly worse on few cases. Motivated by these findings, we integrate MLPs with the two multi-scale approaches and propose two novel plug-and-play modules: *multi-patch MLP* and *multi-downsampling MLP*, designed to enhance the temporal representation of input time series.

**Variable Correlation Modeling.** In multivariate time series data, variable relationships are often complex and diverse, with some variables being strongly correlated while others are weakly correlated or unrelated. Properly leveraging these correlations is critical. Existing methods like iTransformer (Liu et al., 2024), Leddam (Yu et al., 2024), Fredformer (Piao et al., 2024), and Crossformer (Zhang & Yan, 2023) employ variable attention to model these relationships effectively. However, variable attention introduces high computational complexity for high-dimensional data and cannot handle the datasets with weak correlations due to substantial noise between variables. Table 1 presents the number of variables and the mean Pearson correlation coefficient (MCC) across all variables for five papular datasets. Appendix A.1 shows the calculation of MCC and demonstrates the inverse relationship between the average correlation and the noise level. Datasets like ECL and Traffic exhibit higher variable correlations with less noise, while ETTm1 and ETTm2 show weaker correlations (0.096 and 0.139), indicating that noise has begun to dominate the variable relationship. Figure 2 (left) compares iTransformer with and without variable attention across datasets with varying correlation strengths. For datasets with higher correlations (e.g., ECL and Traffic), attention mechanisms improve performance. Conversely, for datasets with weak correlations and more noise (e.g., ETTm1 and ETTm2), attention mechanisms reduce performance. Additionally, as shown in Figure 2 (right), the time and memory costs of variable attention grow quadratically with the number of variables, making it inefficient for high-dimensional datasets. To address these limitations, we propose *grouped bidirectional RNNs* (Grouped BiRNNs) for variable correlation modeling. This approach uses a bidirectional RNN to aggregate useful information from other variables and suppress the influence of variable noise from weakly correlated variables. For high-dimensional datasets, we group variables and apply bidirectional RNNs within groups to capture intra-group correlations. Inter-group information is exchanged via the final hidden states of each intra-group RNNs. This grouping strategy significantly reduces memory and time costs while maintaining effective variable correlation modeling.

We conduct extensive experiments to compare TimesMR with eighteen existing models on sixteen datasets. The results demonstrate that TimesMR consistently outperforms existing methods, achieving state-of-the-art performance on most datasets. Summing up, our contributions are as follows:

• We propose a novel model, TimesMR, for multivariate time series forecasting to effectively model temporal dependencies and variable correlations.
• We introduce two multi-scale MLP modules, multi-patch MLP and multi-downsampling MLP, which enhance temporal representation by combining MLPs with multi-scale techniques.
• We propose grouped bidirectional RNNs, an effective and efficient approach for variable correlation modeling, reducing computational costs while maintaining performance.
• Extensive experiments on sixteen datasets demonstrate the superior performance of TimesMR.

## 2 RELATED WORK

We review existing works in time series forecasting from two perspectives: temporal dependency modeling and variable relation modeling, which are the core components of time series forecasting.

**Temporal dependency modeling.** Recent advancements in time series forecasting have demonstrated the potential of various paradigms for capturing temporal dependencies, including Transformer-based (Zhou et al., 2021; Wu et al., 2021; Zhou et al., 2022), CNN-based (Wu et al., 2023a; Wang et al., 2023; Luo & Wang, 2024), RNN-based (Hochreiter & Schmidhuber, 1997; Lai et al., 2018; Lin et al., 2023), and MLP-based (Zeng et al., 2023; Li et al., 2023) models. Among these, multi-scale modeling approaches, such as TimeMixer (Wang et al., 2024), Pathformer (Chen et al., 2024b), TimesNet (Wu et al., 2023a), and MICN (Wang et al., 2023), have gained prominence for their ability to capture complex temporal patterns. TimeMixer and Pathformer, in particular, represent state-of-the-art methods, employing distinct multi-scale division strategies and feature extractors to enhance temporal dependency modeling. However, these methods often involve intricate designs, such as mixing architecture or costly attention mechanisms, which can be computationally expensive to train. In contrast, lightweight MLP-based models like DLinear (Zeng et al., 2023) and RLinear (Li et al., 2023) have shown competitive performance. Building on this, we propose two novel multi-scale MLP modules that are both efficient and effective in capturing temporal dependencies, offering a simpler yet powerful alternative to existing approaches.

**Variable relation modeling.** Exploring variable relationships in multivariate time series forecasting has garnered significant attention, with existing methods categorized into Variable Dependent (Zhou et al., 2021; Wu et al., 2021; Zhou et al., 2022; Wu et al., 2023a; Chen et al., 2024b; Wang et al., 2024; Liu et al., 2022b), Variable Independent (Nie et al., 2023; Zeng et al., 2023; Lin et al., 2023; 2024b;a), Attention-based (Liu et al., 2024; Yu et al., 2024; Zhang & Yan, 2023; Piao et al., 2024; Lin et al., 2025), Cluster-based (Qiu et al., 2025; Chen et al., 2024a), MLP-based (Ekambaram et al., 2023; Han et al., 2024), CNN-based (Luo & Wang, 2024), Mamba-based (Wang et al., 2025b; Ma et al., 2025), and Graph-based (Wu et al., 2020; Hu et al., 2025) approaches. Attention-based methods excel in capturing token dependencies but often incur high computational costs for high-dimensional datasets, while MLP-based methods are efficient but struggle to model complex variable relationships effectively. In this work, we revisit the potential of RNNs for modeling variable relationships and propose grouped bidirectional RNNs. This approach divides variables into multiple groups, enabling efficient intra-group and inter-group correlation modeling, thereby effectively capturing complex relationships with reduced computational overhead.

## 3 PRELIMINARIES

Multivariate time series $\mathcal{X} \in \mathbb{R}^{C \times T}$ is a $C$-dimensional numerical sequence indexed by time, where $T$ is the number of timestamps, and $C$ is the number of variables ($C > 1$). The input time series is represented as a matrix, where each row corresponds to a variable and each column corresponds to a timestamp. The goal of multivariate time series forecasting is to predict future values based on past observations. Specifically, given a historical time series $\mathcal{X}_{t-L+1:t} \in \mathbb{R}^{C \times L}$, where $L$ is the lookback length, the task is to predict the future values $\hat{\mathcal{X}}_{t+1:t+H} \in \mathbb{R}^{C \times H}$, where $H$ is the prediction length:

$$\hat{\mathcal{X}}_{t+1:t+H} = f(\mathcal{X}_{t-L+1:t}), \text{ where } f \text{ is a forecasting function.} \tag{1}$$

The objective is to learn an accurate forecasting function $f$ that minimizes the prediction error between the predicted time series $\hat{\mathcal{X}}_{t+1:t+H}$ and the ground truth $\mathcal{Y}_{t+1:t+H} \in \mathbb{R}^{C \times H}$. The prediction error is quantified using the Mean Squared Error (MSE) loss function: $\mathcal{L} = \left\| \mathcal{Y}_{t+1:t+H} - \hat{\mathcal{X}}_{t+1:t+H} \right\|_2^2$.

Thus, the multivariate time series forecasting task is formulated as a supervised learning problem with input $\mathcal{X}_{t-L+1:t}$ and the output is the predicted time series $\hat{\mathcal{X}}_{t+1:t+H}$.

## 4 THE PROPOSED METHOD TIMESMR

The overall framework of TimesMR is shown in Figure 3(a). Its core components include the multi-scale MLP module, depicted in Figure 3(b), and the variable correlation module with Grouped

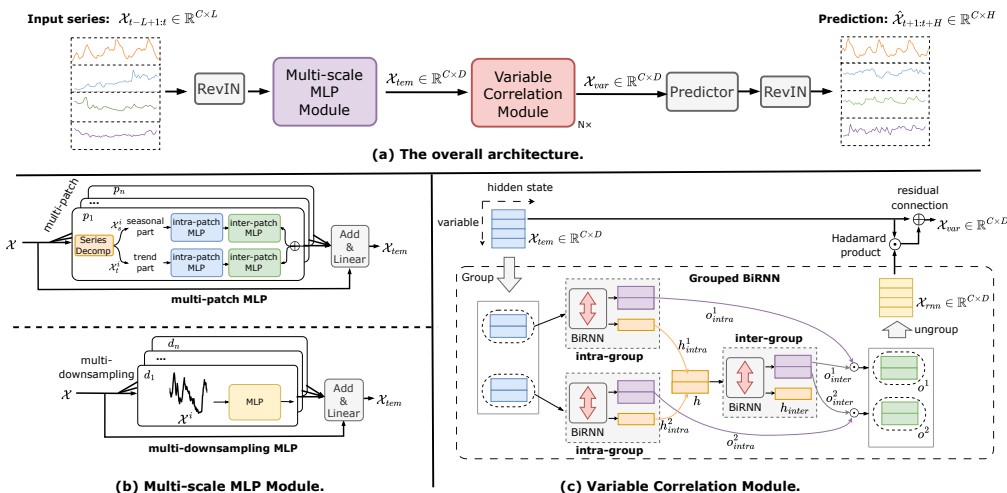

Figure 3: The framework of TimesMR.

BiRNN, shown in Figure 3(c). In Figure 3(a), after passing throught a RevIN module (Kim et al., 2022), the multivariate time series $\mathcal{X}_{t-L+1:t} \in \mathbb{R}^{C \times L}$ is then fed into the multi-scale MLP module, which uses either multi-patch MLP or multi-downsampling MLP to capture temporal dependencies (Section 4.1) and embed the series into a hidden representation $\mathcal{X}_{tem} \in \mathbb{R}^{C \times D}$, where $D$ is the hidden dimension. Subsequently, $\mathcal{X}_{tem}$ is passed through $N$ Variable Correlation module layers with Grouped BiRNNs (Section 4.2) to model complex variable relationships, producing variable representations $\mathcal{X}_{var} \in \mathbb{R}^{C \times D}$. These representations are then used to predict future values $\hat{\mathcal{X}} \in \mathbb{R}^{C \times H}$. Figure 3(b) illustrates the two multi-scale MLP modules. In the multi-patch MLP, the time series is decomposed into seasonal and trend components using patch sizes from $P = \{p_1, p_2, ..., p_n\}$. MLPs are applied to capture intra-patch and inter-patch features, modeling both local and global patterns. The seasonal and trend components are integrated, and the multi-scale features are combined with the input $\mathcal{X}$ using a residual connection. A linear mapping $\text{Linear}(\cdot)$ is then applied to produce the final embedding $\mathcal{X}_{tem} \in \mathbb{R}^{C \times D}$. In the multi-downsampling MLP, the time series is downsampled into multi-scale representations, which are processed by MLPs and similarly combined to produce $\mathcal{X}_{tem}$. Figure 3(c) details the Variable Correlation module with Grouped BiRNNs. The input $\mathcal{X}_{tem}$ is divided into $G$ groups, and BiRNNs are applied within each group to pass intra-group variable information, producing outputs $o_{intra}^j$ and last hidden states $h_{intra}^j$ for each group $j$. The hidden states $h_{intra}^j$ are concatenated and passed through another BiRNN to exchange inter-group information, yielding outputs $o_{inter} \in \mathbb{R}^{G \times D}$. Each group's updated representation $o^j$ is obtained by fusing $o_{inter}^j$ with $o_{intra}^j$ via Hadamard product. The grouped outputs are then ungrouped to form $\mathcal{X}_{rnn} \in \mathbb{R}^{C \times D}$, which is fused with the input $\mathcal{X}_{tem}$ to enhance variable representations. Finally, a residual connection ensures training stability, producing the final variable representations $\mathcal{X}_{var} \in \mathbb{R}^{C \times D}$.

## 4.1 MULTI-SCALE MLP MODULES FOR TEMPORAL DEPENDENCY MODELING

In this section, we present two multi-scale MLP modules for temporal dependency modeling: Multi-patch MLP and Multi-downsampling MLP, as illustrated in Figure 3(b). These modules begin by dividing the input time series into multiple scales. Multi-patch MLP segments the series into patches of varying sizes to capture local and global patterns, while Multi-downsampling MLP progressively reduces the temporal resolution to extract fine-grained details at smaller scales and macroscopic trends at coarser scales. MLPs are then applied to extract features at each scale, leveraging their simplicity and effectiveness in time series forecasting. The features from different scales are fused to form a unified multi-scale representation, which is combined with the original input via a residual connection for stability. Finally, a linear transformation projects the combined features into a high-dimensional space, producing the time series embedding.

**Multi-patch MLP.** For the input multivariate time series $\mathcal{X} \in \mathbb{R}^{C \times L}$, we define $n$ patch sizes $P = \{p_1, p_2, ..., p_n\}$ (e.g., $P = \{1, 4, 12, 24\}$), where each patch size $p_i$ captures features at a specific scale. Using average pooling with kernel size $p_i$, we decompose the time series into seasonal

$\mathcal{X}_s^i$ and trend $\mathcal{X}_t^i$ components for each scale $p_i$. The decomposition process for each $p_i$ is:

$$\mathcal{X}_t^i = \mathrm{Avgpool}(\mathcal{X})_{kernel=p_i}, \tag{2}$$

$$\mathcal{X}_s^i = \mathcal{X} - \mathcal{X}_t^i \tag{3}$$

where $\mathcal{X}_s^i, \mathcal{X}_t^i \in \mathbb{R}^{C \times L}$ represent the seasonal and trend components under patch size $p_i$, respectively. $\mathrm{Avgpool}(\mathcal{X})_{kernel=p_i}$ denotes the average pooling operation with kernel size $p_i$.

Then, for $\mathcal{X}_s^i, \mathcal{X}_t^i$, we reshape them into $\mathcal{X}_s^{i'}, \mathcal{X}_t^{i'} \in \mathbb{R}^{C \times \frac{L}{p_i} \times p_i}$ based on the patch size $p_i$, where $p_i$ is the patch size and $\frac{L}{p_i}$ is the number of patches. This reshaping separates patch information to facilitate the modeling of intra-patch and inter-patch features. To capture the intra-patch features (i.e., local features), we apply MLP along the $p_i$ dimension, obtaining the seasonal and trend representations $\mathcal{X}_{s,intra}^i, \mathcal{X}_{t,intra}^i \in \mathbb{R}^{C \times \frac{L}{p_i} \times p_i}$ under patch size $p_i$:

$$\mathcal{X}_{s,intra}^i = \text{intra-patch MLP}(\mathcal{X}_s^{i'}), \mathcal{X}_{t,intra}^i = \text{intra-patch MLP}(\mathcal{X}_t^{i'}), \tag{4}$$

Then we transpose the last two dimensions of $\mathcal{X}_{s,intra}^i, \mathcal{X}_{t,intra}^i$ and also use MLP to capture the inter-patch features (i.e., global features) along $\frac{L}{p_i}$ dimension:

$$\mathcal{X}_{s,inter}^i = \text{inter-patch MLP}({\mathcal{X}_{s,intra}^i}^{\mathrm{T}}), \mathcal{X}_{t,inter}^i = \text{inter-patch MLP}({\mathcal{X}_{t,intra}^i}^{\mathrm{T}}), \tag{5}$$

where ${\mathcal{X}_{s,intra}^i}^{\mathrm{T}}, {\mathcal{X}_{t,intra}^i}^{\mathrm{T}} \in \mathbb{R}^{C \times p_i \times \frac{L}{p_i}}$ are the transpose of $\mathcal{X}_{s,intra}^i, \mathcal{X}_{t,intra}^i$ and $\mathcal{X}_{s,inter}^i, \mathcal{X}_{t,inter}^i \in \mathbb{R}^{C \times p_i \times \frac{L}{p_i}}$ are the output of inter-patch MLP.

Next, we aggregate the representation of seasonal part and trend part to get the final representation $\mathcal{X}_{out}^i \in \mathbb{R}^{C \times p_i \times \frac{L}{p_i}}$ under patch size $p_i$:

$$\mathcal{X}_{out}^i = \mathcal{X}_{s,inter}^i + \mathcal{X}_{t,inter}^i. \tag{6}$$

We reshape each representation $\mathcal{X}_{out}^i$ into $\mathcal{X}_{out}^i \in \mathbb{R}^{C \times L}$ to align the shape with the original series. Next, we sum the representations across all patch sizes to get the final multi-scale feature representation $\mathcal{X}_m \in \mathbb{R}^{C \times L}$:

$$\mathcal{X}_m = \sum_{i=1}^{n} \mathcal{X}_{out}^i, \tag{7}$$

Finally, we combine the multi-scale features with the input $\mathcal{X}$ using a residual connection, followed by a linear mapping $\mathrm{Linear}(\cdot)$ to transform and project them into the temporal embedding $\mathcal{X}_{tem} \in \mathbb{R}^{C \times D}$, where $D$ is the hidden dimension.

$$\mathcal{X}_{tem} = \mathrm{Linear}(\mathcal{X} + \mathcal{X}_m) \tag{8}$$

**Multi-downsampling MLP.** For the input $\mathcal{X} \in \mathbb{R}^{C \times L}$, we downsample it into a set of multi-scale time series $\mathscr{X} = \{\mathcal{X}^1, \mathcal{X}^2, ..., \mathcal{X}^n\}$ using average pooling with kernel sizes $Down = \{d_1, d_2, ..., d_n\}$ (e.g., $Down = \{1, 4, 12, 24\}$). Each downsampled series $\mathcal{X}^i \in \mathbb{R}^{C \times \frac{L}{d_i}}$ represents the input at a specific granularity, where $\mathcal{X}^1$ captures fine-grained variations and $\mathcal{X}^n$ captures macroscopic trends. For each downsampled series $\mathcal{X}^i$, we apply an MLP along the $\frac{L}{d_i}$ dimension to extract global features, producing a representation $\mathcal{X}_{out}^i \in \mathbb{R}^{C \times L}$. This step also restores the original sequence length, aligning the outputs across scales for fusion:

$$\mathcal{X}_{out}^i = \mathrm{MLP}(\mathcal{X}^i), \tag{9}$$

We aggregate the representations $\mathcal{X}_{out}^i$ from all scales to get the multi-scale features, producing the final representation $\mathcal{X}_m \in \mathbb{R}^{C \times L}$: $\mathcal{X}_m = \sum_{i=1}^{n} \mathcal{X}_{out}^i$. Finally, similar to the Multi-patch MLP, we combine the multi-scale features with the input $\mathcal{X}$ using a residual connection, followed by a linear mapping $\mathrm{Linear}(\cdot)$ to transform and project them into the temporal embedding $\mathcal{X}_{tem} \in \mathbb{R}^{C \times D}$: $\mathcal{X}_{tem} = \mathrm{Linear}(\mathcal{X} + \mathcal{X}_m)$.

## 4.2 GROUPED BiRNN FOR VARIABLE CORRELATION MODELING

After capturing temporal dependencies and getting the embedding $\mathcal{X}_{tem} \in \mathbb{R}^{C \times D}$, the proposed Grouped BiRNN in Variable Correlation module computes the correlation information $\mathcal{X}_{rnn} \in \mathbb{R}^{C \times D}$ from other variables. To optimize BiRNN processing along the variable dimension and accelerate training, variables are grouped, enabling parallel BiRNN operations within each group to capture intra-group interaction. The BiRNN outputs represent intra-group information exchange, while the last hidden states aggregate group-level correlations, facilitating inter-group information passing. As shown in Figure 3, we divide the variables into groups to optimize computation. By default, if the variable number is $C$, we set the number of groups to $G = \lfloor \sqrt{C} \rfloor$. To ensure each group contains $g = \lfloor \frac{C}{G} \rfloor + 1$ variables, we pad $C$ to $C' = G \times g$. This grouping avoids computing pairwise correlations for all variables, which is computationally expensive for high-dimensional datasets. Additionally, as analyzed in Appendix B.4, varying group sizes have minimal impact on performance, demonstrating that Grouped BiRNN effectively captures correlations across all variables.

For each group $\mathcal{X}^j \in \mathbb{R}^{g \times D}, j = \{1, 2, ..., G\}$, we apply a BiRNN along the variable dimension to capture intra-group correlations. This yields the output representations $o_{intra}^j \in \mathbb{R}^{g \times D}$ and the last hidden state $h_{intra}^j \in \mathbb{R}^{1 \times D}$:

$$h_{intra}^j, o_{intra}^j = \text{BiRNN}(\mathcal{X}^j) \tag{10}$$

We concatenate the last hidden states of all groups $h_{intra}^j, j = \{1, 2, ..., G\}$ into $h_{intra} \in \mathbb{R}^{G \times D}$ to pass inter-group correlation information. A BiRNN is then applied to $h_{intra}$, producing the output representations $o_{inter} \in \mathbb{R}^{G \times D}$ and the final hidden state $h_{inter} \in \mathbb{R}^{1 \times D}$:

$$h_{intra} = h_{intra}^1 \parallel h_{intra}^2 \parallel ... \parallel h_{intra}^G, \tag{11}$$

$$h_{inter}, o_{inter} = \text{BiRNN}(h_{intra}) \tag{12}$$

where $\parallel$ denotes Concatenate operation along variable dimension.

The $j$-th element $o_{inter}^j \in \mathbb{R}^{1 \times D}$ in $o_{inter}$ represents the updated representation of the $j$-th group after inter-group information passing. To fuse this updated representation $o_{inter}^j, j = \{1, 2, ..., G\}$ with the corresponding intra-group representation $o_{intra}^j \in \mathbb{R}^{g \times D}$, we use broadcasting and Hadamard product to compute the final correlation representation for the $j$-th group $o^j \in \mathbb{R}^{g \times D}$:

$$o^j = o_{inter}^j \odot o_{intra}^j \tag{13}$$

We complete the information passing for all variables within and across groups. Next, we concatenate the correlation information from all groups $o^j \in \mathbb{R}^{g \times D}, j = \{1, 2, ..., G\}$ along the variable dimension to reconstruct the full correlation representation $\mathcal{X}_{un} \in \mathbb{R}^{C' \times D}$, where $C' = G \times g$.

$$\mathcal{X}_{un} = o^1 \parallel o^2 \parallel ... \parallel o^G, \tag{14}$$

Finally, we truncate $\mathcal{X}_{un}$ to match the original variable count $C$, ensuring dimensional consistency. This yields the final output of Grouped BiRNN, $\mathcal{X}_{rnn} \in \mathbb{R}^{C \times D}$:

$$\mathcal{X}_{rnn} = \text{Truncate}(\mathcal{X}_{un}) \tag{15}$$

## 5 EXPERIMENTS

### 5.1 SETUP

**Datasets.** We evaluate the proposed TimesMR model on sixteen datasets, encompassing diverse temporal patterns and variable relationships. For long-term forecasting (prediction lengths: $\{96, 192, 336, 720\}$), we use nine real-world datasets: ETTh1, ETTh2, ETTm1, ETTm2, ECL, Exchange, Traffic, Weather (from Autoformer (Wu et al., 2021)), and Solar-Energy (from LSTNet (Lai et al., 2018)). For short-term forecasting (prediction lengths: $\{12, 24, 48, 96\}$), we include seven public datasets: PEMS03, PEMS04, PEMS07, PEMS08 (from SCINet (Liu et al., 2022a)), PEMS-BAY, METR-LA (Li et al., 2018), and PEMSD7 (Sen et al., 2019). These datasets provide a comprehensive benchmark for multivariate time series forecasting. Dataset details are in Appendix B.1.

**Baselines.** We evaluate TimesMR against eighteen latest models of diverse variable relationship modeling. These include two variable dependent models: TimeMixer (Wang et al., 2024) and Pathformer (Chen et al., 2024b); five variable attention models: Leddam (Yu et al., 2024), Crossformer (Zhang & Yan, 2023), Fredformer (Piao et al., 2024), iTransformer (Liu et al., 2024), and TQNet (Lin et al., 2025); one variable CNN model: ModernTCN (Luo & Wang, 2024); two variable Mamba models: SMamba (Wang et al., 2025b) and TimePro (Ma et al., 2025); one variable clustering model: DUET (Qiu et al., 2025); two variable MLP models: TSMixer (Ekambaram et al., 2023) and SOFTS (Han et al., 2024); four variable independent models: DLinear (Zeng et al., 2023), PatchTST (Nie et al., 2023), SegRNN (Lin et al., 2023), and CycleNet (Lin et al., 2024a); and one graph-based model: TimeFilter (Hu et al., 2025). As the code of TimeMixer++ (Wang et al., 2025a) has not been released and we are unable to reproduce the results, we exclude it from our comparison. For simplicity, we denote TimesMR with the multi-patch MLP module as TimesMR/patch and TimesMR with the multi-downsampling MLP module as TimesMR/down. The implementation details and hyperparameter sensitivity analysis are provided in Appendix B. In the following, we report the main results and experimental analysis. Additional experimental analysis about the multi-scale MLP module and Variable Correlation module are in Appendix C.

## 5.2 MAIN RESULTS

Tables 2 and 3 summarize the performance of TimesMR and baseline models on all datasets, using Mean Squared Error (MSE) and Mean Absolute Error (MAE) as evaluation metrics. Our proposed TimesMR/patch and TimesMR/down achieve superior results across nearly all datasets and metrics. For instance, on the PEMS07 dataset (883 variables) in Table 2, TimesMR/patch achieves a 35.3% reduction in MSE compared to the variable attention model iTransformer, a 38.1% reduction compared to the variable CNN model ModernTCN, a 16.0% reduction compared to the variable MLP model SOFTS, a 35.3% reduction compared to the variable independent model CycleNet, and a 31.1% reduction compared to the variable dependent model Pathformer. Meanwhile, TimesMR also demonstrates significant improvements on datasets with a relatively small variable number, such as Solar-Energy dataset (137 variables). Specifically, TimesMR/down achieves a 18.4% reduction in MSE compared to iTransformer, a 10.0% reduction compared to ModernTCN, a 17.0% reduction compared to SOFTS, a 9.5% reduction compared to CycleNet, and a 4.0% reduction compared to Pathformer. Overall, TimesMR achieves state-of-the-art performance across diverse datasets, demonstrating its effectiveness in capturing temporal dependencies through multiscale MLP modeling and in modeling variable relationships through the grouped BiRNN approach. We also provide Additional prediction showcases are provided in Appendix G.

Table 2: Full results on short-term prediction datasets. The input length $L$ is fixed as 96 and the results are averaged from all prediction horizons of $H \in \{12, 24, 48, 96\}$. The results of other models are sourced from iTransformer (Liu et al., 2024) and SOFTS (Han et al., 2024). The best results are highlighted in **bold** and the second best are underlined. $1^{st}$ denote the count of the best performance.

| Models | PEMS03 | | PEMS04 | | PEMS07 | | PEMS08 | | PEMSD7 | | PEMS-BAY | | METR-LA | | $1^{st}$ |
|---|---|---|---|---|---|---|---|---|---|---|---|---|---|---|---|
| | MSE | MAE | MSE | MAE | MSE | MAE | MSE | MAE | MSE | MAE | MSE | MAE | MSE | MAE | |
| TimeMixer(2024) | 0.123 | 0.240 | 0.154 | 0.264 | 0.120 | 0.255 | 0.149 | 0.272 | 0.377 | 0.364 | 0.416 | 0.331 | 0.742 | 0.496 | 0 |
| Pathformer(2024b) | 0.104 | 0.214 | 0.100 | 0.213 | 0.106 | 0.210 | 0.151 | 0.239 | 0.359 | 0.342 | 0.393 | 0.317 | 0.776 | 0.490 | 0 |
| DLinear(2023) | 0.262 | 0.356 | 0.260 | 0.351 | 0.310 | 0.373 | 0.341 | 0.372 | 0.593 | 0.474 | 0.709 | 0.443 | 0.765 | 0.521 | 0 |
| PatchTST(2023) | 0.150 | 0.260 | 0.171 | 0.283 | 0.165 | 0.267 | 0.242 | 0.285 | 0.485 | 0.412 | 0.596 | 0.382 | 0.762 | 0.488 | 0 |
| SegRNN(2023) | 0.135 | 0.239 | 0.147 | 0.255 | 0.136 | 0.232 | 0.157 | 0.277 | 0.447 | 0.388 | 0.487 | 0.341 | 0.720 | 0.474 | 0 |
| CycleNet/MLP(2024a) | 0.118 | 0.225 | 0.119 | 0.231 | 0.113 | 0.189 | 0.150 | 0.228 | 0.503 | 0.429 | 0.604 | 0.397 | 0.748 | 0.503 | 0 |
| SOFTS(2024) | 0.104 | 0.210 | 0.102 | 0.208 | 0.087 | 0.184 | 0.138 | 0.219 | 0.503 | 0.429 | 0.604 | 0.397 | 0.748 | 0.503 | 0 |
| TSMixer(2023) | 0.158 | 0.263 | 0.129 | 0.247 | 0.117 | 0.221 | 0.201 | 0.294 | 0.488 | 0.394 | 0.523 | 0.336 | 0.833 | 0.490 | 0 |
| ModernTCN(2024) | 0.111 | 0.226 | 0.103 | 0.217 | 0.118 | 0.208 | 0.230 | 0.268 | 0.374 | 0.370 | 0.436 | 0.330 | 0.808 | 0.500 | 0 |
| SMamba(2025b) | 0.122 | 0.228 | 0.103 | 0.211 | 0.089 | 0.188 | 0.148 | 0.224 | 0.387 | 0.363 | 0.426 | 0.301 | 0.790 | 0.489 | 0 |
| TimePro(2025) | 0.119 | 0.225 | 0.114 | 0.226 | 0.096 | 0.200 | 0.252 | 0.259 | 0.412 | 0.370 | 0.577 | 0.389 | 0.787 | 0.533 | 0 |
| DUET(2025) | 0.110 | 0.219 | 0.113 | 0.227 | 0.091 | 0.198 | 0.130 | 0.229 | 0.360 | 0.336 | 0.447 | 0.326 | 0.813 | 0.487 | 0 |
| Leddam(2024) | 0.119 | 0.224 | 0.132 | 0.240 | 0.152 | 0.247 | 0.146 | 0.223 | 0.354 | 0.340 | 0.517 | 0.357 | 0.738 | 0.495 | 1 |
| Crossformer(2023) | 0.106 | 0.217 | 0.106 | 0.222 | 0.107 | 0.211 | 0.225 | 0.254 | 0.330 | 0.322 | 0.375 | 0.281 | 0.753 | 0.486 | 0 |
| Fredformer(2024) | 0.122 | 0.237 | 0.129 | 0.241 | 0.113 | 0.226 | 0.147 | 0.250 | 0.388 | 0.357 | 0.437 | 0.316 | 0.767 | 0.518 | 0 |
| iTransformer(2024) | 0.113 | 0.221 | 0.119 | 0.231 | 0.113 | 0.189 | 0.150 | 0.228 | 0.382 | 0.354 | 0.447 | 0.319 | 0.837 | 0.536 | 0 |
| TQNet(2025) | 0.097 | 0.203 | 0.091 | 0.197 | 0.075 | 0.171 | 0.142 | 0.229 | 0.361 | 0.343 | 0.427 | 0.315 | 0.749 | 0.489 | 0 |
| TimeFilter(2025) | 0.108 | 0.217 | 0.114 | 0.226 | 0.101 | 0.207 | 0.139 | 0.230 | 0.498 | 0.404 | 0.547 | 0.358 | 0.819 | 0.493 | 0 |
| **TimesMR/patch** | 0.097 | **0.197** | **0.084** | **0.186** | **0.073** | **0.171** | 0.108 | **0.198** | **0.322** | **0.307** | **0.365** | **0.273** | 0.735 | 0.460 | **10** |
| **TimesMR/down** | **0.096** | 0.202 | 0.085 | 0.188 | 0.075 | 0.174 | **0.105** | **0.198** | 0.323 | 0.308 | 0.367 | 0.275 | **0.731** | **0.459** | 5 |

Table 3: Full results on long-term prediction datasets. The input length $L$ is fixed as 96 and the results are averaged from all prediction horizons of $H \in \{96, 192, 336, 720\}$.

| Models | ETTm1 MSE | ETTm1 MAE | ETTm2 MSE | ETTm2 MAE | ETTh1 MSE | ETTh1 MAE | ETTh2 MSE | ETTh2 MAE | Electricity MSE | Electricity MAE | Exchange MSE | Exchange MAE | Traffic MSE | Traffic MAE | Weather MSE | Weather MAE | Solar-Energy MSE | Solar-Energy MAE | $1^{st}$ |
|---|---|---|---|---|---|---|---|---|---|---|---|---|---|---|---|---|---|---|---|
| TimeMixer(2024) | 0.381 | 0.395 | 0.275 | 0.323 | 0.447 | 0.440 | 0.364 | 0.395 | 0.182 | 0.272 | 0.391 | 0.453 | 0.484 | 0.297 | 0.240 | 0.271 | 0.216 | 0.280 | 0 |
| Pathformer(2024b) | 0.407 | 0.412 | 0.281 | 0.323 | 0.465 | 0.445 | 0.389 | 0.407 | 0.188 | 0.281 | 0.427 | 0.437 | 0.512 | 0.332 | 0.240 | 0.271 | 0.198 | 0.271 | 0 |
| DLinear(2023) | 0.403 | 0.407 | 0.350 | 0.401 | 0.456 | 0.452 | 0.559 | 0.515 | 0.212 | 0.300 | 0.354 | 0.414 | 0.625 | 0.383 | 0.265 | 0.317 | 0.330 | 0.401 | 0 |
| PatchTST(2023) | 0.396 | 0.406 | 0.287 | 0.330 | 0.453 | 0.446 | 0.385 | 0.410 | 0.189 | 0.276 | 0.398 | 0.438 | 0.454 | 0.286 | 0.256 | 0.279 | 0.236 | 0.266 | 0 |
| SegRNN(2023) | 0.386 | 0.405 | 0.321 | 0.343 | **0.419** | 0.432 | 0.390 | 0.410 | 0.185 | 0.277 | 0.439 | 0.439 | 0.660 | 0.324 | 0.249 | 0.291 | 0.217 | 0.262 | 1 |
| CycleNet/MLP(2024a) | 0.379 | 0.396 | **0.266** | **0.314** | 0.457 | 0.441 | 0.388 | 0.409 | 0.168 | 0.259 | 0.372 | 0.407 | 0.472 | 0.301 | 0.243 | 0.271 | 0.210 | 0.261 | 2 |
| SOFTS(2024) | 0.393 | 0.403 | 0.287 | 0.330 | 0.449 | 0.442 | 0.373 | 0.400 | 0.174 | 0.264 | 0.401 | 0.427 | 0.409 | **0.267** | 0.255 | 0.279 | 0.229 | 0.256 | 2 |
| TSMixer(2023) | 0.398 | 0.407 | 0.289 | 0.333 | 0.463 | 0.452 | 0.401 | 0.417 | 0.186 | 0.287 | 0.375 | 0.412 | 0.522 | 0.357 | 0.256 | 0.279 | 0.260 | 0.297 | 0 |
| ModernTCN(2024) | 0.386 | 0.401 | 0.278 | 0.322 | 0.445 | 0.432 | 0.381 | 0.404 | 0.197 | 0.282 | 0.406 | 0.427 | 0.546 | 0.348 | 0.240 | 0.271 | 0.211 | 0.310 | 0 |
| DUET(2025) | 0.393 | 0.399 | 0.285 | 0.329 | 0.445 | 0.437 | 0.378 | 0.404 | 0.170 | 0.265 | 0.361 | 0.405 | 0.469 | 0.289 | 0.249 | 0.274 | 0.220 | 0.263 | 0 |
| SMamba(2025b) | 0.398 | 0.405 | 0.288 | 0.332 | 0.455 | 0.450 | 0.381 | 0.405 | 0.170 | 0.265 | 0.367 | 0.408 | 0.414 | 0.276 | 0.251 | 0.276 | 0.240 | 0.273 | 0 |
| TimePro(2025) | 0.391 | 0.400 | 0.281 | 0.326 | 0.438 | 0.438 | 0.377 | 0.403 | 0.169 | 0.262 | 0.352 | 0.399 | 0.443 | 0.287 | 0.251 | 0.276 | 0.232 | 0.266 | 0 |
| Leddam(2024) | 0.386 | 0.397 | 0.281 | 0.325 | 0.431 | 0.429 | 0.373 | 0.399 | 0.169 | 0.263 | 0.367 | 0.404 | 0.467 | 0.297 | 0.242 | 0.272 | 0.230 | 0.264 | 1 |
| Crossformer(2023) | 0.513 | 0.496 | 0.757 | 0.610 | 0.653 | 0.621 | 0.942 | 0.684 | 0.244 | 0.334 | 0.940 | 0.707 | 0.550 | 0.304 | 0.259 | 0.315 | 0.641 | 0.639 | 0 |
| Fredformer(2024) | 0.384 | 0.395 | 0.279 | 0.324 | 0.435 | 0.426 | 0.365 | 0.393 | 0.175 | 0.269 | 0.408 | 0.422 | 0.431 | 0.287 | 0.246 | 0.272 | 0.226 | 0.261 | 0 |
| iTransformer(2024) | 0.407 | 0.410 | 0.288 | 0.332 | 0.454 | 0.447 | 0.383 | 0.407 | 0.178 | 0.270 | 0.360 | 0.403 | 0.428 | 0.282 | 0.258 | 0.278 | 0.233 | 0.262 | 0 |
| TQNet(2025) | 0.377 | 0.393 | 0.277 | 0.323 | 0.441 | 0.434 | 0.378 | 0.402 | 0.164 | 0.259 | 0.368 | 0.405 | 0.445 | 0.276 | 0.242 | 0.269 | 0.198 | 0.256 | 0 |
| TimeFilter(2025) | 0.377 | 0.393 | 0.272 | 0.321 | 0.420 | **0.428** | 0.364 | 0.397 | **0.158** | 0.256 | 0.381 | 0.413 | **0.407** | 0.268 | **0.239** | 0.269 | 0.223 | 0.250 | 4 |
| **TimesMR/patch** | **0.375** | **0.386** | 0.273 | 0.316 | 0.440 | 0.430 | 0.362 | **0.392** | 0.161 | **0.255** | **0.353** | **0.398** | 0.426 | 0.280 | **0.239** | **0.266** | **0.190** | **0.239** | **10** |
| TimesMR/down | 0.377 | 0.389 | 0.269 | 0.316 | 0.436 | 0.430 | **0.358** | **0.392** | 0.161 | **0.255** | 0.359 | 0.402 | 0.429 | 0.280 | **0.239** | 0.268 | 0.194 | 0.244 | 4 |

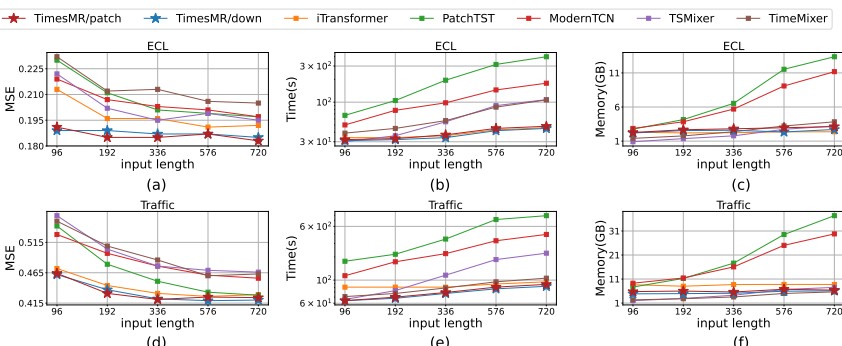

Figure 4: Effectiveness and efficiency under different input length.

## 5.3 MODEL ANALYSIS

**Effectiveness and Efficiency When Varying Input Length.** To demonstrate the effectiveness and efficiency of our TimesMR in capturing temporal dependency and variable relationship, we conduct experiments with different input lengths in {96, 192, 336, 576, 720}. On ECL dataset, Figures 4 (a,b,c) report the MSE, training time per epoch in log with base 10 and memory consumption of TimesMR/patch and TimesMR/down and baselines under different input lengths, while Figure 4 (d,e,f) show the results on Traffic dataset. On both datasets, we observe that under different input lengths, our TimesMR/patch and TimesMR/down consistently outperform all other models in terms of MSE, indicating that our methods can effectively capture temporal dependency and variable relationship (Figures 4 (a) and (d)). Moreover, TimesMR/patch and TimesMR/down are more efficient than other models in terms of training time (Figures 4 (b,c)), while having moderate memory consumption (slightly higher than variable mlp and variable dependent methods) (Figures 4 (e,f)). The results demonstrate the effectiveness and efficiency of the proposed techniques in Section 4.1 that uses lightweight MLPs to capture temporal dependency and Section 4.2 that uses grouped BiRNN to model variable relationship. We provide more analysis results of TimesMR in exploring variable relationship under larger variable number on different datasets in Appendix D.

**Analysis of Each Module.** We evaluate the effectiveness of each module in TimesMR by integrating them into other models. Specifically, we incorporate the Grouped BiRNN module for modeling variable relationships into various baselines and present the results in Table 4. Similarly, the impact of integrating the Multi-patch MLP and Multi-downsampling MLP modules into other models is detailed in Table 13 in Appendix C. For a fair comparison, we reproduce the baseline results and evaluate the addition of Grouped BiRNN under consistent hyperparameter settings. The results indicate that models lacking explicit mechanisms for modeling variable relationships, such as variable-

Table 4: The performance of plugging in the Grouped BiRNN module into other models. **imp(%)** denotes the relative improvement.

| Datasets | | ECL | | Traffic | | Solar-Energy | | PEMS03 | | PEMS04 | | PEMS07 | | PEMS08 | | PEMS-BAY | |
|---|---|---|---|---|---|---|---|---|---|---|---|---|---|---|---|---|---|
| Metric | | MSE | MAE | MSE | MAE | MSE | MAE | MSE | MAE | MSE | MAE | MSE | MAE | MSE | MAE | MSE | MAE |
| Pathformer | avg | 0.188 | 0.280 | 0.512 | 0.332 | 0.198 | 0.271 | 0.104 | 0.214 | 0.100 | 0.213 | 0.105 | 0.209 | 0.151 | 0.239 | 0.393 | 0.317 |
| + G-BiRNN | avg | 0.187 | 0.280 | 0.479 | 0.316 | 0.205 | 0.281 | 0.099 | 0.209 | 0.098 | 0.210 | 0.096 | 0.199 | 0.136 | 0.231 | 0.423 | 0.303 |
| | imp(%) | **0.5** | **0** | **6.4** | **4.8** | -3.5 | -3.7 | **4.8** | **2.3** | **2.0** | **1.4** | **8.6** | **4.8** | **9.9** | **3.3** | -7.6 | **4.4** |
| PatchTST | avg | 0.188 | 0.279 | 0.509 | 0.305 | 0.221 | 0.272 | 0.150 | 0.260 | 0.171 | 0.283 | 0.165 | 0.266 | 0.243 | 0.285 | 0.597 | 0.383 |
| + G-BiRNN | avg | 0.175 | 0.274 | 0.499 | 0.297 | 0.204 | 0.253 | 0.114 | 0.232 | 0.120 | 0.239 | 0.115 | 0.232 | 0.193 | 0.252 | 0.444 | 0.308 |
| | imp(%) | **6.9** | **1.8** | **2.0** | **2.6** | **7.7** | **7.0** | **24.0** | **10.8** | **29.8** | **15.5** | **30.3** | **12.8** | **20.5** | **11.6** | **25.6** | **19.6** |
| iTransformer | avg | 0.178 | 0.267 | 0.440 | 0.294 | 0.217 | 0.267 | 0.121 | 0.229 | 0.126 | 0.238 | 0.106 | 0.210 | 0.245 | 0.272 | 0.448 | 0.320 |
| + G-BiRNN | avg | 0.174 | 0.266 | 0.437 | 0.290 | 0.210 | 0.269 | 0.112 | 0.222 | 0.107 | 0.219 | 0.107 | 0.212 | 0.210 | 0.254 | 0.440 | 0.315 |
| | imp(%) | **2.2** | **0.4** | **0.7** | **1.4** | **3.2** | -0.7 | **7.4** | **3.0** | **15.1** | **8.0** | -0.9 | -0.9 | **14.3** | **6.6** | **1.8** | **1.6** |
| TSMixer | avg | 0.192 | 0.292 | 0.539 | 0.347 | 0.336 | 0.340 | 0.158 | 0.263 | 0.129 | 0.247 | 0.117 | 0.221 | 0.201 | 0.294 | 0.523 | 0.337 |
| + G-BiRNN | avg | 0.184 | 0.284 | 0.523 | 0.338 | 0.217 | 0.299 | 0.111 | 0.228 | 0.100 | 0.216 | 0.086 | 0.190 | 0.128 | 0.235 | 0.423 | 0.316 |
| | imp(%) | **4.2** | **2.7** | **3.0** | **2.6** | **35.4** | **12.0** | **29.7** | **13.3** | **22.5** | **12.5** | **26.5** | **14.0** | **36.3** | **20.1** | **19.1** | **6.2** |
| ModernTCN | avg | 0.192 | 0.296 | 0.520 | 0.338 | 0.211 | 0.310 | 0.111 | 0.226 | 0.103 | 0.217 | 0.126 | 0.221 | 0.230 | 0.268 | 0.436 | 0.330 |
| + G-BiRNN | avg | 0.192 | 0.295 | 0.515 | 0.335 | 0.224 | 0.322 | 0.105 | 0.214 | 0.098 | 0.210 | 0.118 | 0.208 | 0.210 | 0.250 | 0.414 | 0.318 |
| | imp(%) | **0** | **0.3** | **1.0** | **0.9** | -6.1 | -3.8 | **5.4** | **5.3** | **4.8** | **3.2** | **6.3** | **5.9** | **8.7** | **6.7** | **5.0** | **3.6** |

Table 5: Ablation analysis of TimesMR/patch and TimesMR/down.

| Datasets | ECL | | Traffic | | Solar-Energy | | PEMS03 | | PEMS04 | | PEMS07 | | PEMS08 | | PEMS-BAY | |
|---|---|---|---|---|---|---|---|---|---|---|---|---|---|---|---|---|
| Metric | MSE | MAE | MSE | MAE | MSE | MAE | MSE | MAE | MSE | MAE | MSE | MAE | MSE | MAE | MSE | MAE |
| **TimesMR/patch** | **0.161** | **0.255** | **0.426** | **0.280** | **0.190** | **0.239** | **0.097** | **0.197** | **0.084** | **0.186** | **0.073** | **0.171** | **0.108** | **0.198** | **0.365** | **0.273** |
| w/o VCM | 0.177 | 0.268 | 0.471 | 0.301 | 0.197 | 0.255 | 0.102 | 0.208 | 0.094 | 0.202 | 0.079 | 0.177 | 0.154 | 0.226 | 0.391 | 0.284 |
| **TimesMR/down** | **0.161** | **0.255** | **0.429** | **0.280** | **0.194** | **0.244** | **0.096** | **0.202** | **0.085** | **0.188** | **0.075** | **0.174** | **0.105** | **0.198** | **0.367** | **0.275** |
| w/o VCM | 0.177 | 0.268 | 0.473 | 0.303 | 0.197 | 0.254 | 0.104 | 0.209 | 0.097 | 0.206 | 0.083 | 0.182 | 0.169 | 0.226 | 0.401 | 0.287 |
| w/o patch or down | 0.167 | 0.262 | 0.432 | 0.285 | 0.193 | 0.261 | 0.097 | 0.200 | 0.085 | 0.188 | 0.080 | 0.184 | 0.109 | 0.202 | 0.376 | 0.281 |

dependent Pathformer and variable-independent PatchTST, benefit substantially from the inclusion of Grouped BiRNN. Furthermore, even models with existing mechanisms for variable relationships, such as attention-based iTransformer, MLP-based TSMixer, and CNN-based models ModernTCN, show consistent performance improvements across nearly all datasets. These findings highlight the versatility of Grouped BiRNN as a plug-and-play module that effectively exchange information among variables, thereby enhancing their representations. A similar trend is observed for the Multi-patch MLP and Multi-downsampling MLP modules, as shown in Table 13 in Appendix C, further demonstrating their utility in boosting the performance of existing models.

**Ablation Study.** We ablate the two main components of TimesMR: the Variable Correlation Module (VCM) in Section 4.2 and and the Multi-scale MLP Modules (patch or down) in Section 4.1. Table 5 shows the performance of TimesMR/patch and TimesMR/down with and without these components. When removing the VCM, we observe a significant performance drop in both models TimesMR/patch and TimesMR/down, indicating that the VCM is crucial for capturing variable relationships. The last row of Table 5 shows the performance of TimesMR/patch and TimesMR/down without either the multi-patch or multi-downsampling MLP modules, indicating that the multi-scale MLP modules are essential for capturing temporal dependencies.

# 6 CONCLUSION

In this paper, we introduced TimesMR, a novel model for multivariate time series forecasting that effectively addresses the challenges of temporal dependency and variable correlation modeling. Our key contributions include: (1) two lightweight multi-scale MLP modules that efficiently capture temporal patterns, and (2) grouped bidirectional RNNs that model variable correlations while reducing computational complexity. Extensive experiments on sixteen datasets demonstrate that TimesMR consistently outperforms eighteen latest baselines in both accuracy and efficiency across most cases. Ablation studies confirm that our proposed modules function effectively as plug-and-play components that can enhance existing architectures. This work reveals the potential of combining simple yet powerful techniques for multivariate time series forecasting, establishing a direction for developing computationally efficient models that scale well to high-dimensional data with broad practical applications. In the future, we will focus on optimizing memory efficiency of Grouped BiRNN.

## REPRODUCIBILITY STATEMENT

To ensure reproducibility, we provide the datasets details and experimental details in Appendix B.

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

## A   ANALYSIS OF INTER-VARIABLE FEATURES

### A.1   THE CALCULATION OF MCC AND THE NOISE LEVEL

As discussed in our Introduction, in multivariate time series data, variable relationships are complex and diverse, with some variables being strongly correlated while others are weakly correlated or unrelated. While the attention mechanism is known for its strong representational power, it also has notable drawbacks as shown in Figure 2 in our Introduction: on certain datasets, it may introduce noise from weakly correlated or uncorrelated variables, and it incurs a high computational cost when the number of variables is large.

To clarify further, we provide a clear, quantifiable inverse relationship between the mean Pearson correlation coefficient of multivariate time series (denoted as MCC in our Introduction) and the noise level. The smaller the average correlation, the more likely it is that the observed correlation between variables is caused by noise. Specifically, we provide the following proof:

Consider two time series $\mathcal{X}$ and $\mathcal{Y}$ :

$$\mathcal{X}_t = \mathcal{S}_{\mathcal{X},t} + \sqrt{\gamma_X} \cdot \mathcal{N}_{\mathcal{X},t} \tag{16}$$

$$\mathcal{Y}_t = a \cdot \mathcal{S}_{\mathcal{X},t} + \mathcal{S}_{\mathcal{Y},t} + \sqrt{\gamma_{\mathcal{Y}}} \cdot \mathcal{N}_{\mathcal{Y},t} \tag{17}$$

where $\mathcal{S}_{\mathcal{X},t}$ is the shared signal component and $\mathcal{S}_{\mathcal{Y},t}$ represents the signal specific to $\mathcal{Y}_t$. $\mathcal{N}_{X,t}$ and $\mathcal{N}_{\mathcal{Y},t}$ represent standard Gaussian noise, and $\gamma_{\mathcal{X}}$ and $\gamma_{\mathcal{Y}}$ are noise intensity parameters.

Then, the correlation without noise is:

$$\rho_{\text{true}} = \frac{\text{cov}(\mathcal{S}_{\mathcal{X}}, a\mathcal{S}_{\mathcal{X}} + \mathcal{S}_{\mathcal{Y}})}{\sigma_{\mathcal{S}_{\mathcal{X}}} \sigma_{a\mathcal{S}_{\mathcal{X}} + \mathcal{S}_{\mathcal{Y}}}} \tag{18}$$

And the correlation with noise is:

$$\rho_{\mathcal{X}\mathcal{Y}} = \frac{\text{cov}(\mathcal{X}, \mathcal{Y})}{\sigma_{\mathcal{X}} \sigma_{\mathcal{Y}}} \tag{19}$$

$$= \frac{\text{cov}(\mathcal{S}_{\mathcal{X}} + \mathcal{N}_{\mathcal{X}}, a\mathcal{S}_{\mathcal{X}} + \mathcal{S}_{\mathcal{Y}} + \mathcal{N}_{\mathcal{Y}})}{\sigma_{\mathcal{X}} \sigma_{\mathcal{Y}}} \tag{20}$$

$$= \frac{\text{cov}(\mathcal{S}_{\mathcal{X}}, a\mathcal{S}_{\mathcal{X}} + \mathcal{S}_{\mathcal{Y}})}{\sigma_{\mathcal{X}} \sigma_{\mathcal{Y}}} \quad \text{(noises are uncorrelated with signals)} \tag{21}$$

$$= \underbrace{\frac{\text{cov}(\mathcal{S}_{\mathcal{X}}, a\mathcal{S}_{\mathcal{X}} + \mathcal{S}_{\mathcal{Y}})}{\sigma_{\mathcal{S}_{\mathcal{X}}} \sigma_{a\mathcal{S}_{\mathcal{X}} + \mathcal{S}_{\mathcal{Y}}}}}_{\rho_{\text{true}}} \cdot \frac{\sigma_{\mathcal{S}_{\mathcal{X}}} \sigma_{a\mathcal{S}_{\mathcal{X}} + \mathcal{S}_{\mathcal{Y}}}}{\sigma_{\mathcal{X}} \sigma_{\mathcal{Y}}} \tag{22}$$

$$= \rho_{\text{true}} \cdot \frac{1}{\sqrt{\left(1 + \frac{\sigma^2_{\mathcal{N}_{\mathcal{X}}}}{\sigma^2_{\mathcal{S}_{\mathcal{X}}}}\right)\left(1 + \frac{\sigma^2_{\mathcal{N}_{\mathcal{Y}}}}{\sigma^2_{a\mathcal{S}_{\mathcal{X}} + \mathcal{S}_{\mathcal{Y}}}}\right)}} \tag{23}$$

$$= \rho_{\text{true}} \cdot \frac{1}{\sqrt{(1 + \frac{1}{\text{SNR}_{\mathcal{X}}})(1 + \frac{1}{\text{SNR}_{\mathcal{Y}}})}} \tag{24}$$

where $\text{SNR}_{\mathcal{X}} = \frac{\sigma^2_{\mathcal{S}_{\mathcal{X}}}}{\gamma_{\mathcal{X}}}$, $\text{SNR}_{\mathcal{Y}} = \frac{\sigma^2_{a\mathcal{S}_{\mathcal{X}} + \mathcal{S}_{\mathcal{Y}}}}{\gamma_{\mathcal{Y}}}$ are the signal-to-noise ratio (SNR) for $\mathcal{X}$ and $\mathcal{Y}$ and $\rho_{\text{true}}$ is the true correlation coefficient without noise contamination. When assuming all variables have similar SNRs, the average correlation for the multi-variables simplifies to:

$$\bar{\rho} = \bar{\rho}_{\text{true}} \cdot \frac{1}{1 + \frac{1}{\text{SNR}}} \tag{25}$$

This equation directly demonstrates the inverse relationship between average correlation and noise level between variables. Particularly, if we set $\bar{\rho}_{\text{true}}$ to 0.5 according to Cohen's effect size guidelines, when $noise\ level = \frac{1}{\text{SNR}} = 1.5$ (corresponding to $SNR \approx 0.67$), the average correlation drops below 0.2, indicating that noise has begun to dominate the variable. At this point, observed correlations between variables are likely caused by noise rather than real variable relationships. Therefore, in

Table 6: The mean Pearson correlation coefficient of all datasets and performance comparison of iTransformer and TimesMR.

| Datasets | | ETTm1 | ETTm2 | ETTh1 | ETTh2 | Weather | Exchange | ECL | Traffic |
|---|---|---|---|---|---|---|---|---|---|
| MCC | | 0.096 | 0.139 | 0.096 | 0.139 | 0.140 | 0.224 | 0.243 | 0.281 |
| iTransformer | MSE | 0.407 | 0.288 | 0.454 | 0.383 | 0.258 | 0.360 | 0.178 | 0.428 |
| TimesMR/patch | MSE | **0.375** | **0.273** | **0.440** | **0.362** | **0.239** | **0.353** | **0.161** | **0.426** |
| Datasets | | PEMS03 | PEMS04 | PEMS07 | PEMS08 | PEMSD7 | PEMS-BAY | METR-LA | Solar-Energy |
| MCC | | 0.418 | 0.383 | 0.399 | 0.389 | 0.206 | 0.179 | 0.308 | 0.454 |
| iTransformer | MSE | 0.113 | 0.119 | 0.113 | 0.150 | 0.382 | 0.447 | 0.837 | 0.233 |
| TimesMR/patch | MSE | **0.097** | **0.084** | **0.073** | **0.108** | **0.322** | **0.365** | **0.735** | **0.190** |

time series analysis, average correlation serves as a reliable indicator of data quality: when $\bar{\rho} < 0.2$ , the correlation may be highly affected by noise. Here, $\bar{\rho}$ refers to the MCC value reported in Table 1 of the Introduction. We observe that the variable correlations in the ETTm1 ($\bar{\rho} = 0.096$), ETTm2 ($\bar{\rho} = 0.139$), and Weather ($\bar{\rho} = 0.140$) datasets are significantly weakened by noise. This suggests that, for certain datasets, suppressing correlation noise between variables is crucial for effective modeling.

Meanwhile, RNN architectures such as LSTM and GRU are particularly effective at suppressing noise and reinforcing clean historical information, thanks to its gating mechanism. For instance, the forget gate controls the retention of past information and can emphasize useful historical information in the presence of noise. Similarly, the input gate can regulate the influence of new (potentially noisy) inputs, thereby mitigating noise contamination. So we introduce the traditional RNN structure from a novel perspective: rather than explicitly computing all variable correlations, we implicitly suppress the influence of weakly correlated variables through RNN structure. This allows the model to retain the useful information from related variables and avoid the interference from weakly related or unrelated variables.

Moreover, we calculate the mean Pearson correlation coefficient (MCC) for all datasets and summarize the performance of iTransformer and TimesMR in Table 6. The results show that on datasets with low MCC values ($MCC < 0.2$), such as ETTm1, ETTm2, ETTh1, ETTh2, Weather, Exchange, and PEMS-BAY, our TimesMR consistently outperforms iTransformer. Furthermore, on datasets with relatively high MCC values, such as PEMS03, PEMS04, PEMS07 and PEMS08, TimesMR still achieves better performance than iTransformer. This indicates that the RNN-based design of TimesMR not only preserves useful correlations among variables to enhance representations, but also effectively suppresses the impact of variable noise in scenarios where correlations are weak.

## A.2 THE ANALYSIS OF PARTIAL CORRELATION AND SPECTRAL COHERENCE

In addition to the dataset-level mean Pearson correlation (MCC) analysis presented, we further analyze the noise characteristics in multivariate time series datasets using partial correlations and spectral coherence, with visualization results reported in Figure 5 and Figure 6 to provide stronger diagnostics.

Specifically, Partial correlation measures the direct linear relationship between two variables while controlling for the influence of all other variables. This allows us to better identify the true pairwise dependencies that are not confounded by other correlated variables. When the number of variables is large, the distribution of partial correlations tends to be biased toward lower values, as controlling for many other variables often reduces the residual linear dependency between a pair of variables. Therefore, in our implementation, for each variable, we select the top-10 most strongly correlated variables as control variables rather than using all other variables. This strategy focuses on the most relevant confounding factors, reduces computational cost in high-dimensional settings, and avoids diluting the residual signal with weakly correlated variables. As a result, the partial correlations accurately reflect the meaningful direct dependencies between variable pairs. Spectral coherence quantifies the degree of linear correlation between two time series at different frequencies. It helps capture frequency-dependent dependencies and lagged relationships, providing a complementary view of the interactions that cannot be observed with simple time domain correlations.

From Figure 5 and Figure 6, we observe that the partial correlations in most datasets are centered around zero, indicating that the direct linear dependencies between variable pairs are generally weak. Similarly, the spectral coherence for most datasets is also close to zero, suggesting that frequency-dependent relationships and lagged correlations are limited. This implies that these datasets contain a relatively high level of noise, which requires the model to accurately capture the meaningful variable relationships while avoiding the influence of weakly related or unrelated variables.

The new findings from partial correlations and spectral coherence, together with the original MCC analysis, further validate our motivation for designing the Grouped BiRNN module to aggregate useful information from other variables and suppress the influence of variable noise from weakly correlated variables.

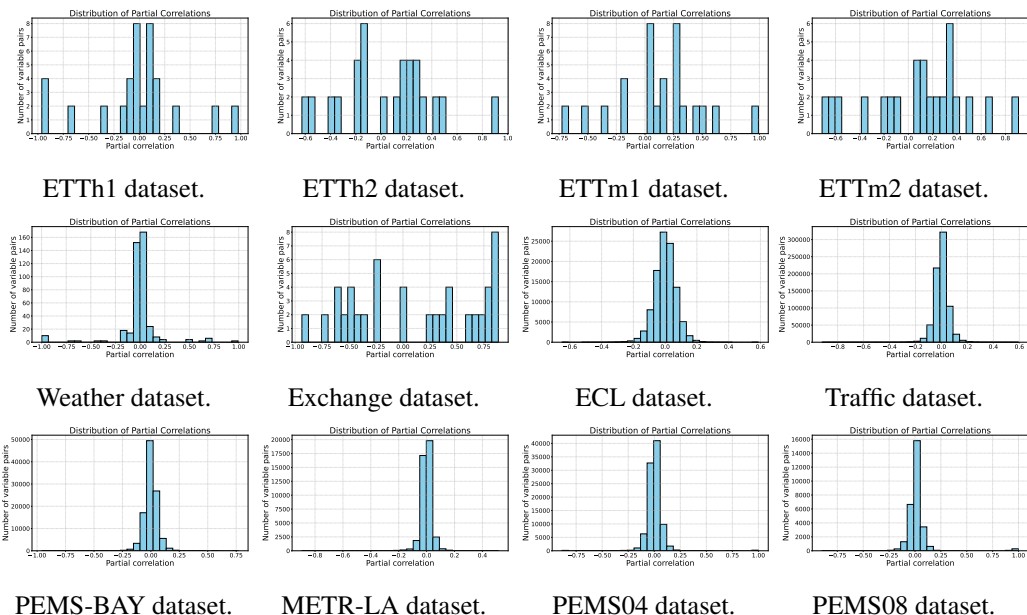

Figure 5: The partial correlation distribution of each dataset.

# B   EXPERIMENTAL DETAILS

## B.1   DATASET DETAILS

The details of the experimental datasets are summarized as follows: (1) *ETT* (Zhou et al., 2021): Contains two subsets, ETTh and ETTm, collected from electricity transformers every 15 minutes and 1 hour, spanning July 2016 to July 2018. (2) *ECL*[1] (Wu et al., 2021): Records hourly electricity consumption of 321 customers from 2012 to 2014. (3) *Exchange* (Lai et al., 2018): Daily exchange rates of eight countries from 1990 to 2016. (4) *Traffic*[2] (Wu et al., 2021): Hourly road occupancy rates from sensors on San Francisco Bay area freeways, collected by the California Department of Transportation. (5) *Weather*[3] (Wu et al., 2021): 21 meteorological indicators recorded every 10 minutes throughout 2020. (6) *Solar-Energy* (Lai et al., 2018): Solar power production of 137 PV plants in 2006, sampled every 10 minutes. (7) *PEMS* (Liu et al., 2022a): Public traffic network data from California, collected in 5-minute intervals, including subsets PEMS03, PEMS04, PEMS07, and PEMS08. (8) *PeMSD7(M)* (Chen et al., 2001): Data from the Caltrain PeMS system, containing 228 time-series collected at 5-minute intervals. (9) *PEMS-BAY* (Li et al., 2018): Traffic data from 325 sensors in the Bay Area, collected over 6 months (Jan 1, 2017, to May 31, 2017) by CalTrans Performance Measurement System (PeMS). (10) *METR-LA* (Jagadish et al., 2014): Traffic data from

---

[1] https://archive.ics.uci.edu/ml/datasets/ElectricityLoadDiagrams20112014
[2] http://pems.dot.ca.gov
[3] https://www.bgc-jena.mpg.de/wetter/

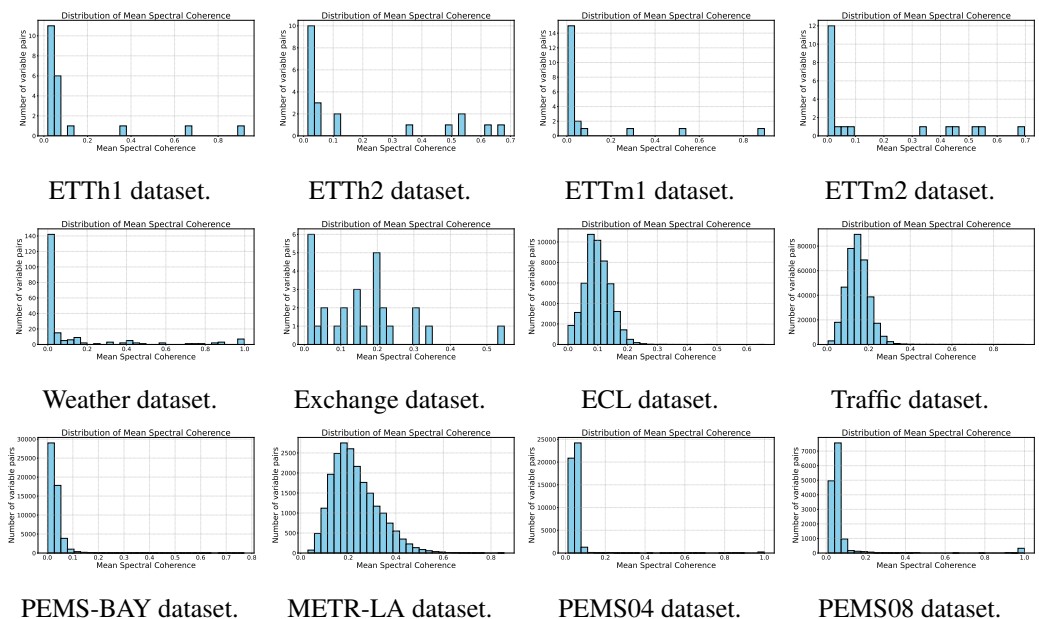

Figure 6: The spectral coherence distribution of each dataset.

207 sensors on Los Angeles County highways, collected over 4 months (Mar 1, 2012, to Jun 30, 2012).

We adopt the same data processing and train-validation-test split protocol from iTransformer (Liu et al., 2024). For long-term forecasting, the lookback length is fixed at 96 for the ETT, ECL, Exchange, Traffic, Weather, and Solar-Energy datasets, with prediction lengths of {96, 192, 336, 720}. For short-term forecasting, the lookback length is also fixed at 96 for the PEMS03, PEMS04, PEMS07, PEMS08, PeMSD7(M), PEMS-BAY, and METR-LA datasets, with prediction lengths of {12, 24, 48, 96}. Detailed dataset statistics are provided in Table 7.

## B.2 IMPLEMENTATION DETAILS

All the experiments are implemented in PyTorch (Paszke et al., 2019) and conducted on NVIDIA A800-SXM4-40GB GPU. We use L2Loss and ADAM optimizer (Kingma & Ba, 2015) with an initial learning rate of 0.001 or 0.0001. Batch size is uniformly set to 32. The training epoch number is set to 15 and training process is early stopped after three epochs if there is no loss degradation on the valid set. The mean square error (MSE) and mean absolute error (MAE) as evaluation metrics. We set the Variable Correlation Module layer $N \in \{1, 2, 4\}$ and the hidden dimension $D$ to 512. We set the multi-patch sizes to $P = \{1, 4, 12, 24\}$ in Multi-patch MLP and downsampling sizes to $Down = \{1, 4, 12, 24\}$ in Multi-downsampling MLP by default. The group number in Grouped BiRNN is set to the square root of the variable number by default and also can be selected from $\{5, 10, 20\}$. We use the results of all baselines reported in iTransformer (Liu et al., 2024) and SOFTS (Han et al., 2024) and their original papers. And for baselines without reported results, we run their publicly available code using default parameter settings. The detailed model configuration information is presented in Table 8. More hyperparameter sensitivity analysis can be seen in the Appendix B.4.

## B.3 ERROR BARS

To verify the stability, we repeat the experiment three times with different random seeds and report the standard deviation in Table 9 and Table 10, which exhibits that the performance of TimesMR is stable.

Table 7: Detailed dataset descriptions. *Dim* denotes the variable number of each dataset. *Dataset Size* denotes the total number of time points in (Train, Validation, Test) split respectively. *Prediction Length* denotes the prediction length in each dataset. *Frequency* denotes the sampling interval of time points.

| Dataset | Dim | Prediction Length | Dataset Size | Frequency | Information |
|---|---|---|---|---|---|
| ETTh1, ETTh2 | 7 | {96, 192, 336, 720} | (8545, 2881, 2881) | Hourly | Electricity |
| ETTm1, ETTm2 | 7 | {96, 192, 336, 720} | (34465, 11521, 11521) | 15min | Electricity |
| Weather | 21 | {96, 192, 336, 720} | (36792, 5271, 10540) | 10min | Weather |
| ECL | 321 | {96, 192, 336, 720} | (18317, 2633, 5261) | Hourly | Electricity |
| Exchange | 8 | {96, 192, 336, 720} | (5120, 665, 1422) | Daily | Economy |
| Traffic | 862 | {96, 192, 336, 720} | (12185, 1757, 3509) | Hourly | Transportation |
| Solar-Energy | 137 | {96, 192, 336, 720} | (36601, 5161, 10417) | 10min | Energy |
| PEMS03 | 358 | {12, 24, 48, 96} | (15617, 5135, 5135) | 5min | Transportation |
| PEMS04 | 307 | {12, 24, 48, 96} | (10172, 3375, 3375) | 5min | Transportation |
| PEMS07 | 883 | {12, 24, 48, 96} | (16911, 5622, 5622) | 5min | Transportation |
| PEMS08 | 170 | {12, 24, 48, 96} | (10690, 3548, 3548) | 5min | Transportation |
| PeMSD7(M) | 228 | {12, 24, 48, 96} | (8763, 1256, 2523) | 5min | Transportation |
| PEMS-BAY | 325 | {12, 24, 48, 96} | (36374, 5201, 10412) | 5min | Transportation |
| METR-LA | 207 | {12, 24, 48, 96} | (23883, 3417, 6843) | 5min | Transportation |

Table 8: Experiment configuration of TimesMR. All the experiments use the ADAM (Kingma & Ba, 2015) optimizer with the default hyperparameter configuration for $(\beta_1, \beta_2)$ as (0.9, 0.999). N denote Variable Correlation Module layer and $sqrt$ denote the the square root of the variable number. $P$ /$Down$ denote the multi-scale sizes in Multi-scale MLP module.

| Datasets/Configurations | N | $P$ /$Down$ | G | D | learning rate | batch size |
|---|---|---|---|---|---|---|
| ETTm1 | 2 | {1, 4, 12, 24} | $sqrt$ | 512 | 0.001 | 32 |
| ETTm2 | 2 | {1, 4, 12, 24} | $sqrt$ | 512 | 0.0001 | 32 |
| ETTh1 | 2 | {1, 4, 12, 24} | $sqrt$ | 512 | 0.001 | 32 |
| ETTh2 | 2 | {1, 4, 12, 24} | $sqrt$ | 512 | 0.001 | 32 |
| Weather | 2 | {1, 4, 12, 24} | $sqrt$ | 512 | 0.001 | 32 |
| ECL | 4 | {1, 4, 12, 24} | 5 | 512 | 0.001 | 32 |
| Exchange | 2 | {1, 4, 12, 24} | $sqrt$ | 512 | 0.0001 | 32 |
| Traffic | 4 | {1, 4, 12, 24} | 5 | 512 | 0.001 | 32 |
| Solar-Energy | 1 | {1, 4, 12, 24} | 20 | 512 | 0.001 | 32 |
| PEMS03 | 2 | {1, 4, 12, 24} | $sqrt$ | 512 | 0.001 | 32 |
| PEMS04 | 2 | {1, 4, 12, 24} | $sqrt$ | 512 | 0.001 | 32 |
| PEMS07 | 2 | {1, 4, 12, 24} | 5 | 512 | 0.001 | 32 |
| PEMS08 | 2 | {1, 4, 12, 24} | 5 | 512 | 0.001 | 32 |
| PeMSD7(M) | 2 | {1, 4, 12, 24} | $sqrt$ | 512 | 0.001 | 32 |
| PEMS-BAY | 2 | {1, 4, 12, 24} | $sqrt$ | 512 | 0.001 | 32 |
| METR-LA | 2 | {1, 4, 12, 24} | $sqrt$ | 512 | 0.001 | 32 |

Table 9: Robustness of TimesMR/patch. The results are obtained from three random seeds.

| Dataset | ETTm1 | | ETTm2 | | ETTh1 | | ETTh2 | |
|---|---|---|---|---|---|---|---|---|
| Horizon | MSE | MAE | MSE | MAE | MSE | MAE | MSE | MAE |
| 96 | 0.303±0.000 | 0.344±0.000 | 0.168±0.000 | 0.250±0.000 | 0.375±0.001 | 0.388±0.001 | 0.276±0.000 | 0.330±0.000 |
| 192 | 0.357±0.000 | 0.374±0.000 | 0.235±0.000 | 0.294±0.000 | 0.429±0.001 | 0.418±0.000 | 0.355±0.001 | 0.379±0.000 |
| 336 | 0.389±0.004 | 0.396±0.002 | 0.296±0.001 | 0.333±0.001 | 0.470±0.000 | 0.439±0.000 | 0.409±0.002 | 0.421±0.000 |
| 720 | 0.453±0.001 | 0.433±0.000 | 0.394±0.000 | 0.390±0.000 | 0.488±0.001 | 0.474±0.000 | 0.409±0.000 | 0.432±0.000 |
| Dataset | ECL | | Traffic | | Weather | | Solar-Energy | |
| Horizon | MSE | MAE | MSE | MAE | MSE | MAE | MSE | MAE |
| 96 | 0.134±0.000 | 0.229±0.000 | 0.390±0.000 | 0.262±0.000 | 0.153±0.000 | 0.196±0.000 | 0.157±0.001 | 0.202±0.001 |
| 192 | 0.153±0.000 | 0.246±0.000 | 0.413±0.000 | 0.273±0.000 | 0.200±0.000 | 0.246±0.000 | 0.196±0.001 | 0.231±0.001 |
| 336 | 0.167±0.000 | 0.263±0.000 | 0.434±0.001 | 0.282±0.000 | 0.260±0.000 | 0.287±0.000 | 0.201±0.001 | 0.260±0.001 |
| 720 | 0.191±0.001 | 0.285±0.001 | 0.470±0.000 | 0.301±0.000 | 0.344±0.000 | 0.342±0.001 | 0.208±0.001 | 0.264±0.000 |

Table 10: Robustness of TimesMR/down.

| Dataset | ETTm1 | | ETTm2 | | ETTh1 | | ETTh2 | |
|---|---|---|---|---|---|---|---|---|
| Horizon | MSE | MAE | MSE | MAE | MSE | MAE | MSE | MAE |
| 96 | 0.304±0.000 | 0.345±0.000 | 0.166±0.000 | 0.250±0.000 | 0.374±0.001 | 0.390±0.001 | 0.279±0.000 | 0.337±0.000 |
| 192 | 0.355±0.000 | 0.375±0.000 | 0.231±0.000 | 0.292±0.000 | 0.427±0.002 | 0.420±0.004 | 0.350±0.004 | 0.377±0.001 |
| 336 | 0.393±0.000 | 0.399±0.000 | 0.291±0.001 | 0.332±0.001 | 0.466±0.003 | 0.442±0.003 | 0.392±0.000 | 0.417±0.000 |
| 720 | 0.456±0.000 | 0.437±0.000 | 0.391±0.000 | 0.389±0.000 | 0.478±0.007 | 0.469±0.007 | 0.410±0.001 | 0.436±0.000 |
| Dataset | ECL | | Traffic | | Weather | | Solar-Energy | |
| Horizon | MSE | MAE | MSE | MAE | MSE | MAE | MSE | MAE |
| 96 | 0.134±0.000 | 0.230±0.000 | 0.390±0.001 | 0.263±0.001 | 0.154±0.000 | 0.199±0.000 | 0.169±0.001 | 0.206±0.000 |
| 192 | 0.152±0.001 | 0.246±0.000 | 0.415±0.001 | 0.274±0.002 | 0.202±0.000 | 0.244±0.000 | 0.194±0.002 | 0.247±0.002 |
| 336 | 0.166±0.000 | 0.261±0.000 | 0.438±0.001 | 0.283±0.000 | 0.261±0.000 | 0.288±0.000 | 0.205±0.000 | 0.261±0.001 |
| 720 | 0.188±0.001 | 0.283±0.000 | 0.473±0.000 | 0.301±0.000 | 0.342±0.000 | 0.342±0.001 | 0.207±0.000 | 0.265±0.001 |

## B.4 HYPERPARAMETER SENSITIVITY

**Group number** As shown in Figure 7, we conduct hyperparameter sensitivity about the group number to analyze the impact of the number of groups in the Grouped BiRNN module. To be more comprehensive, we expand the experiments to more datasets with moderate and large variable sizes: PEMS03 (358 variables), PEMS04 (307 variables), PEMS07 (883 variables), PEMS08 (170 variables), Traffic (862 variables), ECL (321 variables) and Solar-Energy (137 variables). We can observe that although training without grouping often yields slightly better forecasting accuracy, it comes at the cost of significantly higher training overhead. In contrast, properly increasing the number of groups effectively reduces training time while maintaining comparable accuracy. This efficiency gain is attributed to two factors: our grouping strategy reduces the number of variables each RNN needs to process within a group; independent groups can be processed in parallel. Considering the trade-off between performance, training efficiency, and ease of hyperparameter selection for new datasets, we choose to use $\sqrt{C}$ as a practical default setting for the group number.

**Patch sizes and downsampling sizes** The choice of patch sizes and downsampling rates can be guided by the sampling frequency of the time series data. For example, the sampling frequency of Traffic dataset is one hour, so we can empirically set the patch sizes and the downsampling rates to [1(one hour), 4(four hours), 12(half day), 24(one day)]. For the ETTm1 dataset whose sampling interval is 15 minutes, we also empirically set the patch sizes and the downsampling rates to [1 (15 minutes), 4 (one hour), 12 (3 hours), 24 (half day)]. This selection is consistent with the practice used in previous papers, such as PatchTST (Nie et al., 2023), Pathformer (Chen et al., 2024b), and TimeMixer (Wang et al., 2024). In this paper, to maintain the applicability of our model, we set the patch sizes and the downsampling rates uniformly across all datasets to $\{1, 4, 12, 24\}$, which can also be directly applied to other datasets. As shown in Figure 8, we evaluate the hyperparameter sensitivity about the patch size in Multi-patch. The results indicate that using multiple patch size effectively captures diverse patterns, demonstrating the effectiveness of multi-scale modeling in capturing temporal dependencies.

We also evaluate the hyperparameter sensitivity of TimesMR/patch with respect to the following factors: Variable Correlation Module layers $N$ and the hidden dimension $D$ in Figure 8. We can conclude that an excessive number of Variable Correlation Module layers or an overly large

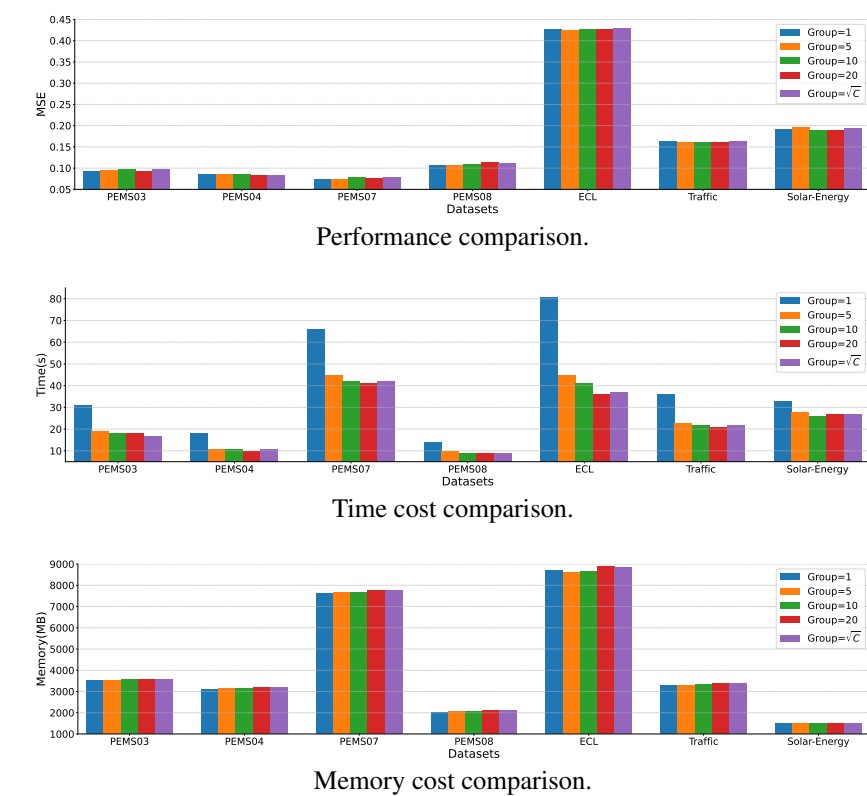

Performance comparison.

Time cost comparison.

Memory cost comparison.

Figure 7: The performance and training cost under different groups.

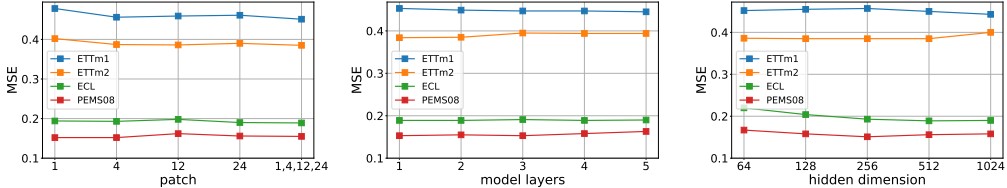

Figure 8: Hyperparameter sensitivity.

hidden dimension can hinder model convergence, emphasizing the need for proper selection of these parameters.

## C  MODULE ANALYSIS

### C.1  MULTI-SCALE MLP MODULE

In this section, we evaluate the effectiveness and efficiency of our Multi-scale MLP module. For clarity, we denote our multi-patch MLP as **Patch** and our multi-downsampling MLP as **Down**.

**Performance Comparison with Temporal Models.**  We compare the performance of our Multi-scale MLP module (**Patch** and **Down**) with several state-of-the-art temporal models, including two recent multi-scale models (Pathformer and TimeMixer), one general-purpose model (TimesNet), and two MLP-based models (TSMixer and DLinear). The results, summarized in Table 11, demonstrate that our simple MLP-based approach outperforms all other temporal models. This highlights that modeling temporal dependencies in time series does not necessarily require complex architectures. Instead, our straightforward multi-scale division combined with MLP achieves superior performance.

Table 11: Performance comparison of our Multi-scale MLP modules and other models.

| Datasets | ECL | | Traffic | | Solar-Energy | | PEMS03 | | PEMS04 | | PEMS07 | | PEMS08 | | PEMS-BAY | |
|---|---|---|---|---|---|---|---|---|---|---|---|---|---|---|---|---|
| Metric | MSE | MAE | MSE | MAE | MSE | MAE | MSE | MAE | MSE | MAE | MSE | MAE | MSE | MAE | MSE | MAE |
| **Patch** | **0.177** | **0.268** | **0.471** | 0.301 | **0.197** | 0.255 | **0.102** | **0.208** | **0.094** | **0.202** | **0.079** | **0.177** | 0.154 | **0.226** | **0.391** | **0.284** |
| **Down** | **0.177** | **0.268** | 0.473 | 0.303 | **0.197** | **0.254** | 0.104 | 0.209 | 0.097 | 0.206 | 0.083 | 0.182 | 0.169 | **0.226** | 0.401 | 0.287 |
| Pathformer | 0.188 | 0.280 | 0.512 | 0.332 | 0.198 | 0.271 | 0.104 | 0.214 | 0.100 | 0.213 | 0.105 | 0.209 | 0.151 | 0.239 | 0.393 | 0.317 |
| TimeMixer | 0.182 | 0.272 | 0.484 | **0.294** | 0.210 | 0.266 | 0.123 | 0.240 | 0.154 | 0.264 | 0.120 | 0.255 | **0.149** | 0.272 | 0.416 | 0.331 |
| TimesNet | 0.192 | 0.295 | 0.620 | 0.336 | 0.301 | 0.319 | 0.147 | 0.248 | 0.129 | 0.241 | 0.124 | 0.225 | 0.193 | 0.271 | 0.684 | 0.384 |
| TSMixer | 0.192 | 0.292 | 0.539 | 0.347 | 0.336 | 0.340 | 0.158 | 0.263 | 0.129 | 0.247 | 0.117 | 0.221 | 0.201 | 0.294 | 0.523 | 0.337 |
| DLinear | 0.212 | 0.300 | 0.625 | 0.383 | 0.330 | 0.401 | 0.262 | 0.356 | 0.260 | 0.351 | 0.310 | 0.373 | 0.341 | 0.372 | 0.709 | 0.443 |

**Efficiency analysis.** We evaluate the efficiency of our proposed Multi-scale MLP module by comparing it with two recent multi-scale modeling methods, Pathformer and TimeMixer, in terms of time and memory cost per epoch. For a fair comparison, we uniformly set the model layers to 2, patch sizes to {4, 12, 24}, and downsampling layers to 3. As shown in Table 12, Pathformer incurs high training time and memory costs due to its intra-patch and inter-patch attention mechanisms. In contrast, our multi-patch MLP approach, which relies solely on simple MLPs to capture intra-patch and inter-patch features, is significantly more efficient. Similarly, our multi-downsampling MLP approach, which uses simple MLPs to model overall changes, demonstrates lower time and memory costs compared to TimeMixer. These results highlight the efficiency of our Multi-scale MLP module (both Patch and Down) in temporal dependency modeling, demonstrating that simpler designs can achieve superior efficiency without sacrificing performance.

Table 12: Efficiency Comparison of our Multi-scale MLP modules and other multi-scale models on different datasets.

| Datasets | ECL | | Traffic | | Solar-Energy | | PEMS03 | | PEMS04 | | PEMS07 | | PEMS08 | | PEMS-BAY | |
|---|---|---|---|---|---|---|---|---|---|---|---|---|---|---|---|---|
| Metric | time | memory | time | memory | time | memory | time | memory | time | memory | time | memory | time | memory | time | memor |
| **Patch** | 36 | 1942 | 40 | 2496 | 48 | 1098 | 23 | 1932 | 15 | 1718 | 35 | 4088 | 16 | 1148 | 28 | 1642 |
| Pathformer | 121 | 8902 | 212 | 11694 | 220 | 4242 | 108 | 9630 | 60 | 8378 | 258 | 23028 | 63 | 4830 | 576 | 8654 |
| **Down** | 20 | 720 | 31 | 822 | 20 | 578 | 15 | 596 | 10 | 578 | 11 | 732 | 10 | 534 | 17 | 730 |
| TimeMixer | 29 | 1040 | 45 | 1206 | 56 | 726 | 19 | 636 | 12 | 604 | 19 | 818 | 17 | 568 | 40 | 812 |

**Multi-scale MLP module on other models.** To further validate the effectiveness of our Multi-scale MLP modules, we integrate them as plug-and-play components into three representative models: PatchTST (variable independent), iTransformer (variable attention-based), and TSMixer (variable MLP-based). For a fair comparison, we reproduce the baseline results and evaluate the performance after incorporating our Multi-scale MLP modules under identical hyperparameter settings. The results, summarized in Table 13, demonstrate that both the Patch and Down modules significantly enhance the performance of all selected models. Notably, for TSMixer, our modules achieve an overall improvement of 10%-40% across all datasets, highlighting their ability to effectively capture temporal dependencies. These findings confirm that our Multi-scale MLP modules can serve as versatile plug-and-play enhancements to improve temporal modeling in diverse architectures.

**The difference of two techniques in Multi-Scale MLP** For clear comparison of the multi-patch MLP and multi-downsampling MLP method, we plot the fragments of some datasets with different characteristics and report the performance of them on these datasets in Figure 9. The Multi-patch MLP is designed to effectively capture local temporal variations by segmenting the input sequence into non-overlapping patches and extracting intra-patch features through MLPs. This patching operation limits the receptive field to short temporal windows, allowing the model to focus on fine-grained, high-frequency patterns such as local fluctuations or short-term changes (such as ETTm1 and Traffic datasets). In contrast, the Multi-downsampling MLP targets global or long-term temporal patterns by reducing the temporal resolution through the downsampling operation. This operation aggregates information over larger time spans, thereby emphasizing overall trends and long-term changes in the time series (such as the ETTh2 and METR-LA datasets).

Table 13: The performance of some other models with our multi-patch and multi-downsampling MLP modules. **imp(%)** denotes the performance promotion.

| Datasets | | ECL | | Traffic | | Solar-Energy | | PEMS03 | | PEMS04 | | PEMS07 | | PEMS08 | | PEMS-BAY | |
|---|---|---|---|---|---|---|---|---|---|---|---|---|---|---|---|---|---|
| Metric | | MSE | MAE | MSE | MAE | MSE | MAE | MSE | MAE | MSE | MAE | MSE | MAE | MSE | MAE | MSE | MAE |
| PatchTST | avg | 0.188 | 0.279 | 0.509 | 0.305 | 0.221 | 0.272 | 0.150 | 0.260 | 0.171 | 0.283 | 0.165 | 0.266 | 0.243 | 0.285 | 0.597 | 0.383 |
| + Patch | avg | 0.183 | 0.277 | 0.505 | 0.299 | 0.209 | 0.271 | 0.128 | 0.242 | 0.142 | 0.261 | 0.131 | 0.236 | 0.221 | 0.270 | 0.530 | 0.377 |
|  | imp(%) | 2.6 | 0.8 | 0.8 | 2.0 | 5.4 | 0.4 | 14.7 | 6.1 | 16.9 | 7.8 | 20.6 | 11.3 | 9.0 | 5.3 | 11.2 | 1.6 |
| + Down | avg | 0.183 | 0.278 | 0.509 | 0.300 | 0.211 | 0.274 | 0.128 | 0.229 | 0.141 | 0.253 | 0.129 | 0.235 | 0.225 | 0.277 | 0.544 | 0.378 |
|  | imp(%) | 2.6 | 0.3 | 0 | 1.6 | 4.5 | -0.7 | 14.7 | 11.9 | 17.5 | 10.6 | 21.8 | 11.6 | 7.4 | 2.8 | 8.9 | 1.3 |
| iTransformer | avg | 0.178 | 0.267 | 0.440 | 0.294 | 0.217 | 0.267 | 0.121 | 0.229 | 0.126 | 0.238 | 0.106 | 0.210 | 0.245 | 0.272 | 0.448 | 0.320 |
| + Patch | avg | 0.171 | 0.264 | 0.446 | 0.287 | 0.208 | 0.274 | 0.103 | 0.209 | 0.100 | 0.212 | 0.082 | 0.180 | 0.217 | 0.252 | 0.374 | 0.286 |
|  | imp(%) | 3.9 | 1.1 | -1.3 | 2.4 | 4.1 | 2.6 | 14.9 | 8.7 | 20.6 | 10.9 | 22.6 | 14.3 | 11.4 | 7.3 | 16.5 | 10.6 |
| + Down | avg | 0.169 | 0.261 | 0.444 | 0.288 | 0.195 | 0.246 | 0.105 | 0.212 | 0.104 | 0.215 | 0.084 | 0.185 | 0.230 | 0.262 | 0.397 | 0.292 |
|  | imp(%) | 5.0 | 2.2 | -0.9 | 2.3 | 10.1 | 7.9 | 13.2 | 7.4 | 17.5 | 9.7 | 20.7 | 11.9 | 6.1 | 3.7 | 11.4 | 8.7 |
| TSMixer | avg | 0.192 | 0.292 | 0.539 | 0.347 | 0.336 | 0.340 | 0.158 | 0.263 | 0.129 | 0.247 | 0.117 | 0.221 | 0.201 | 0.294 | 0.523 | 0.337 |
| + Patch | avg | 0.173 | 0.273 | 0.510 | 0.331 | 0.222 | 0.295 | 0.097 | 0.208 | 0.087 | 0.195 | 0.078 | 0.177 | 0.133 | 0.224 | 0.391 | 0.294 |
|  | imp(%) | 9.9 | 6.5 | 5.4 | 4.6 | 33.9 | 13.2 | 38.6 | 20.9 | 32.5 | 21.1 | 33.3 | 19.9 | 33.8 | 23.8 | 25.2 | 12.7 |
| + Down | avg | 0.176 | 0.277 | 0.508 | 0.329 | 0.207 | 0.271 | 0.103 | 0.214 | 0.088 | 0.199 | 0.079 | 0.181 | 0.139 | 0.232 | 0.384 | 0.296 |
|  | imp(%) | 8.3 | 5.1 | 5.7 | 5.2 | 38.3 | 20.3 | 34.8 | 18.6 | 31.8 | 19.4 | 32.4 | 18.1 | 30.8 | 21.1 | 26.6 | 12.2 |

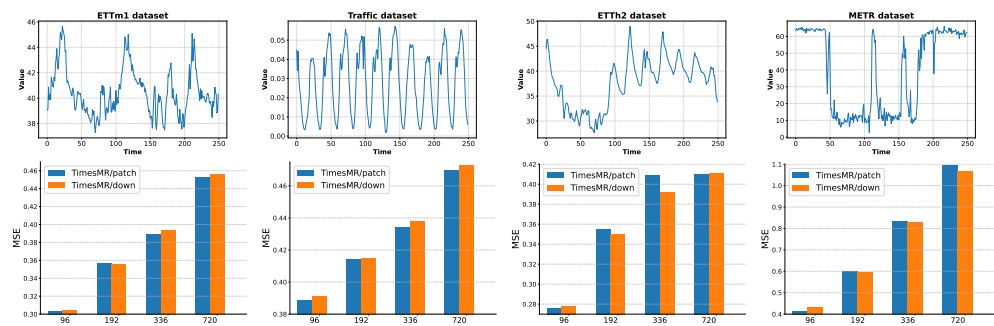

Figure 9: The showcases of different datasets and the performance of multi-patch and multi-downsampling MLP on these datasets.

**Combination of Multi-scale Modules**   We report the performance of combining both the multi-patch and multi-downsampling MLP modules in the Figure 10. The results show that using both modules together does not lead to a significant performance improvement. We attribute this to two main reasons: (1) there is an inherent overlap in their operations, especially when the patch size and downsampling rate are both set to 1, and (2) each module individually already possesses sufficient capacity to capture temporal dependencies effectively. This suggests that either module alone is capable of modeling temporal dependencies to a high degree of accuracy, and their combination introduces slight redundancy rather than complementary gains.

## C.2 VARIABLE CORRELATION MODULE

In this section, we evaluate the effectiveness and efficiency of our Variable Correlation module, specifically focusing on the Grouped BiRNN design. Grouped BiRNN divides variables into smaller groups, enabling efficient correlation modeling while mitigating interference from weakly related or unrelated variables.

**Effectiveness: Grouped BiRNN vs. BiRNN.**   To assess the effectiveness of Grouped BiRNN, we compare its performance with the original BiRNN. As shown in Table 14, Grouped BiRNN achieves comparable results to BiRNN, demonstrating its ability to capture relationships across all variables effectively, even when processed in smaller groups.

**Efficiency: Training Time and Memory.**   We also evaluate the efficiency of Grouped BiRNN by comparing its training time and memory usage with BiRNN. As shown in Table 15, Grouped BiRNN

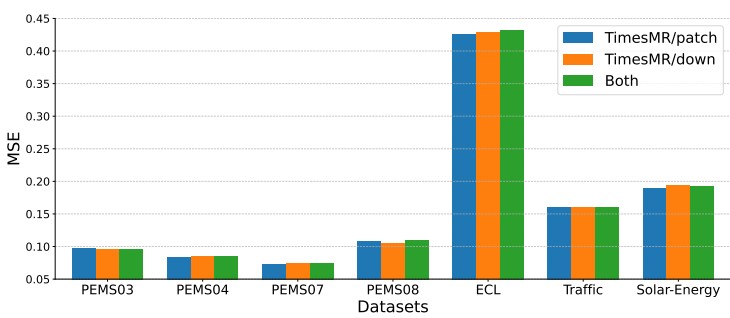

Figure 10: Combination of multi-patch and multi-downsampling MLP modules.

Table 14: The performance comparison of BiRNN and Grouped BiRNN.

| Datasets | | ECL | | Traffic | | Solar-Energy | | PEMS03 | | PEMS04 | | PEMS07 | | PEMS08 | | PEMS-BAY | |
|---|---|---|---|---|---|---|---|---|---|---|---|---|---|---|---|---|---|
| Metric | | MSE | MAE | MSE | MAE | MSE | MAE | MSE | MAE | MSE | MAE | MSE | MAE | MSE | MAE | MSE | MAE |
| TimesMR/patch | avg | 0.161 | 0.255 | 0.426 | 0.280 | 0.190 | 0.239 | 0.097 | 0.197 | 0.084 | 0.186 | 0.073 | 0.171 | 0.108 | 0.198 | 0.365 | 0.273 |
| BiRNN | avg | 0.164 | 0.259 | 0.427 | 0.278 | 0.191 | 0.243 | 0.088 | 0.193 | 0.087 | 0.193 | 0.073 | 0.170 | 0.106 | 0.199 | 0.359 | 0.275 |
| TimesMR/down | avg | 0.161 | 0.255 | 0.429 | 0.280 | 0.194 | 0.244 | 0.096 | 0.202 | 0.085 | 0.188 | 0.075 | 0.174 | 0.105 | 0.198 | 0.367 | 0.275 |
| BiRNN | avg | 0.165 | 0.260 | 0.426 | 0.277 | 0.192 | 0.247 | 0.087 | 0.194 | 0.089 | 0.196 | 0.072 | 0.169 | 0.109 | 0.202 | 0.362 | 0.273 |

significantly reduces training time due to the smaller number of variables processed simultaneously. While Grouped BiRNN incurs slightly higher memory usage due to padding operations along the variable dimension, this trade-off is acceptable given the substantial time savings. Future work will focus on optimizing memory usage further.

**Compared with other variable relationship modeling methods**  To evaluate the effectiveness of our Grouped BiRNN even further, we compare it with other general variable relationship modeling methods: variable attention method and variable MLP method. Specifically, we replace the Grouped BiRNN in our TimesMR/patch model with these other methods and compare the performance. As shown in Figure 11(a), replacing with MLP causes the largest performance drop, and replacing with attention also causes a performance drop, suggesting that attention is indeed superior to MLP in modeling variable relationship. Meanwhile, under the same settings, replacing our Grouped BiRNN with any other method will lead to a performance degradation. This proves that our Grouped BiRNN is far superior to these existing methods in dealing with the complex variable relationship.

Further, to verify that the performance gains arise from the RNN component itself rather than the grouping strategy alone, we combined the grouping approach with different modeling modules: attention and MLP, and compared their performance and training costs. Specifically, to ensure a fair comparison, we implemented both intra-group and inter-group versions of Attention and MLP. The results illustrated in Figure 11(b,c,d) show that, even under the same grouping strategy, our Grouped BiRNN consistently outperforms others across datasets. Regarding training cost, the attention-based model incurs the highest computational overhead due to its complexity, while our RNN-based method's cost is only slightly above that of the MLP approach, demonstrating a favorable balance between efficiency and effectiveness. These findings confirm that the RNN module provides significant benefits beyond the grouping strategy alone.

## D  COMPUTATIONAL COMPLEXITY WITH DIFFERENT VARIABLE NUMBERS

In this paper we employ RNNs specifically to model inter-variable relationships rather than temporal dependencies. Consequently, the challenge we face shifts from computationally inefficient for long sequences to handling high-dimensional variable numbers. An effective approach must balance high performance with computational efficiency. Referring to the original Transformer paper, we provide the computational complexity analysis of our Grouped BiRNN as follows.

The overall complexity primarily arises from two components: the intra-group BiRNN and the inter-group BiRNN. Let the number of variables be $C$, the hidden dimension be $D$, and the number of

Table 15: The time and memory comparison of BiRNN and Grouped BiRNN.

| Datasets | ECL | | Traffic | | Solar-Energy | | PEMS03 | | PEMS04 | | PEMS07 | | PEMS08 | | PEMS-BAY | |
| --- | --- | --- | --- | --- | --- | --- | --- | --- | --- | --- | --- | --- | --- | --- | --- | --- |
| Metric | time | memory | time | memory | time | memory | time | memory | time | memory | time | memory | time | memory | time | memor |
| TimesMR/patch | **21** | 2194 | **66** | 8188 | **19** | 932 | **17** | 2368 | **10** | 2160 | **46** | 5014 | **7** | 1372 | **52** | 2884 |
| BiRNN | **30** | 2130 | **98** | 8082 | **22** | 886 | **27** | 2270 | **15** | 2030 | **70** | 4934 | **10** | 1308 | **84** | 2710 |
| TimesMR/down | **19** | 2112 | **57** | 7278 | **15** | 912 | **16** | 2290 | **9** | 2094 | **43** | 4820 | **7** | 1340 | **50** | 2802 |
| BiRNN | **29** | 2054 | **91** | 7174 | **20** | 846 | **26** | 2194 | **15** | 1968 | **67** | 4734 | **9** | 1276 | **82** | 2630 |

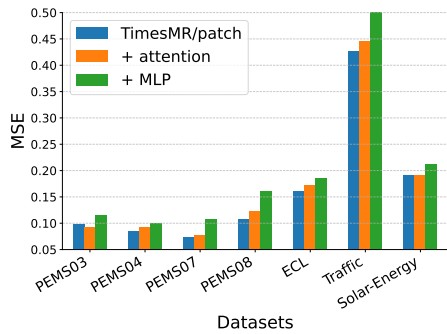

(a) Performance comparison with attention and MLP.

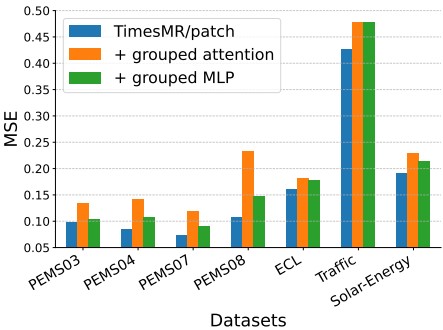

(b) Performance comparison with grouped attention and grouped MLP.

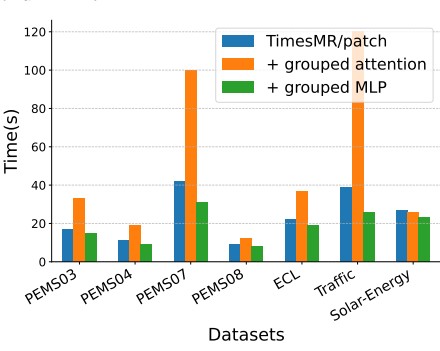

(c) Time cost comparison per epoch with grouped attention and grouped MLP.

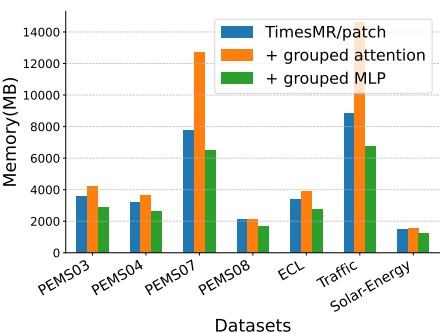

(d) Memory cost comparison per epoch with grouped attention and grouped MLP.

Figure 11: Compared with attention and MLP.

groups be $g$. Each group thus contains $\frac{C}{g}$ variables. The complexity of the intra-group BiRNN for a single group is $\mathcal{O}(2 \cdot \frac{C}{g} \cdot D^2)$. With $g$ groups, the total complexity becomes $\mathcal{O}(2 \cdot g \cdot \frac{C}{g} \cdot D^2)$. However, since different groups can be processed in parallel, the effective complexity of the intra-group BiRNN reduces to $\mathcal{O}(2 \cdot \frac{C}{g} \cdot D^2)$. The inter-group BiRNN, which models interactions across the group-level representations, has a complexity of $\mathcal{O}(2 \cdot g \cdot D^2)$. Therefore, the total complexity of our Grouped BiRNN is $\mathcal{O}(2 \cdot \frac{C}{g} \cdot D^2 + 2 \cdot g \cdot D^2)$. If we set the number of groups $g$ to $\sqrt{C}$ by default, the total complexity becomes $\mathcal{O}(\sqrt{C} \cdot D^2)$, which scales with the square root of the variable number $C$. In this work, we use the GRU variant as our basic RNN method. Since GRU has approximately three times the computational complexity of a standard RNN, the overall complexity still scales with the square root of the variable number $C$. In contrast, iTransformer adopts attention mechanisms to model inter-variable dependencies, incurring a computational complexity of $\mathcal{O}(C^2 \cdot D)$, which scales quadratically with $C$. This makes attention-based models less efficient in high-dimensional settings.

To illustrate this further, we compare our TimesMR with several popular variable relationship modeling methods: variable attention (iTransformer), variable MLP (TSMixer), variable CNN (ModernTCN), variable-independent (PatchTST), and variable-dependent (TimeMixer). We conduct experiments on datasets with varying numbers of variables: Solar (137 variables), ECL (321 vari-

ables), PEMS-BAY (325 variables), PEMS03 (358 variables), Traffic (862 variables), and PEMS07 (883 variables). The time and memory usage results are presented in Figure 12. From the results, we observe that variable MLP method TSMixer and variable-dependent method TimeMixer generally require less memory but fail to capture complex variable relationships effectively, leading to poor performance (see Tables 2 and 3). For datasets with a large number of variables, TimesMR demonstrates lower training costs compared to iTransformer, ModernTCN, and PatchTST, while maintaining similar training time to TSMixer and TimeMixer. Importantly, TimesMR consistently outperforms all other methods in multivariate time series forecasting (see Tables 2 and 3), highlighting its effectiveness and efficiency in modeling variable relationships.

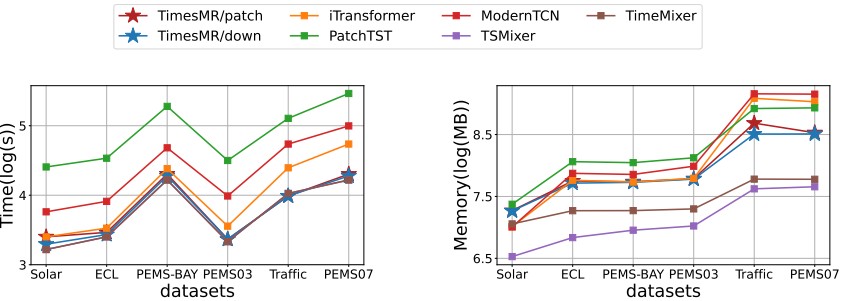

Figure 12: Time and memory comparison of different datasets with different variable numbers.

## E DIFFERENT RNN METHODS

The default RNNs method used in our Grouped BiRNN is GRU (Cho et al., 2014). Here we also compare with RNN and LSTM (Hochreiter & Schmidhuber, 1997) to verify the rationality and superiority of our design. The results are provided in Table 16. We find that the performance of using LSTM and using GRU is about the same and both are better than RNN. This indicates that LSTM and GRU are more capable than RNN in passing correlation information and avoid weakly related or unrelated information interference. Since GRU is more lightweight than LSTM, we use GRU uniformly in this paper.

Table 16: The performance comparison with RNN and LSTM.

| Datasets | | ECL | | Traffic | | Solar-Energy | | PEMS03 | | PEMS04 | | PEMS07 | | PEMS08 | | PEMS-BAY | |
|---|---|---|---|---|---|---|---|---|---|---|---|---|---|---|---|---|---|---|
| Metric | | MSE | MAE | MSE | MAE | MSE | MAE | MSE | MAE | MSE | MAE | MSE | MAE | MSE | MAE | MSE | MAE |
| TimesMR/patch | avg | 0.161 | 0.255 | 0.426 | 0.280 | 0.190 | 0.239 | 0.097 | 0.197 | 0.084 | 0.186 | 0.073 | 0.171 | 0.108 | 0.198 | 0.365 | 0.273 |
| +LSTM | avg | 0.160 | 0.255 | 0.428 | 0.281 | 0.191 | 0.244 | 0.096 | 0.196 | 0.083 | 0.185 | 0.072 | 0.177 | 0.105 | 0.196 | 0.366 | 0.274 |
| +RNN | avg | 0.164 | 0.258 | 0.431 | 0.284 | 0.198 | 0.257 | 0.095 | 0.197 | 0.085 | 0.189 | 0.074 | 0.174 | 0.110 | 0.203 | 0.368 | 0.276 |
| TimesMR/down | avg | 0.161 | 0.255 | 0.429 | 0.280 | 0.194 | 0.244 | 0.096 | 0.202 | 0.085 | 0.188 | 0.075 | 0.174 | 0.105 | 0.198 | 0.367 | 0.275 |
| +LSTM | avg | 0.159 | 0.254 | 0.431 | 0.281 | 0.192 | 0.247 | 0.097 | 0.200 | 0.084 | 0.186 | 0.075 | 0.172 | 0.105 | 0.199 | 0.370 | 0.274 |
| +RNN | avg | 0.162 | 0.257 | 0.430 | 0.283 | 0.196 | 0.251 | 0.097 | 0.201 | 0.086 | 0.191 | 0.075 | 0.175 | 0.106 | 0.200 | 0.364 | 0.276 |

## F THE USE OF LARGE LANGUAGE MODELS (LLMS)

Large Language Models (LLMs) are only used to correct grammatical errors and to polish the syntax of the article.

## G SHOWCASES

To evaluate model performance, we present prediction showcases from the test dataset in Figures 13 and 14. We compare TimesMR with several baselines, including TimePro (Ma et al., 2025), Time-Filter (Hu et al., 2025) CycleNet (Lin et al., 2024a), Fredformer (Zhou et al., 2022), iTransformer (Liu et al., 2024), ModernTCN (Luo & Wang, 2024), Pathformer (Chen et al., 2024b), TimeMixer (Wang et al., 2024), SegRNN (Lin et al., 2023) and PatchTST (Nie et al., 2023), on the ECL and PEMS07

datasets. These baselines employ diverse strategies for modeling temporal dependencies and variable relationships, ensuring a comprehensive comparison.

On the ECL dataset, TimesMR demonstrates superior performance, particularly around the 100th and 270th time points, effectively capturing abrupt pattern changes. On the PEMS07 dataset, TimesMR excels in fitting the overall trend, showcasing its ability to model long-term temporal dynamics. These results highlight TimesMR's robustness in handling both short-term cyclical variations and long-term trend changes, making it well-suited for real-world forecasting scenarios.

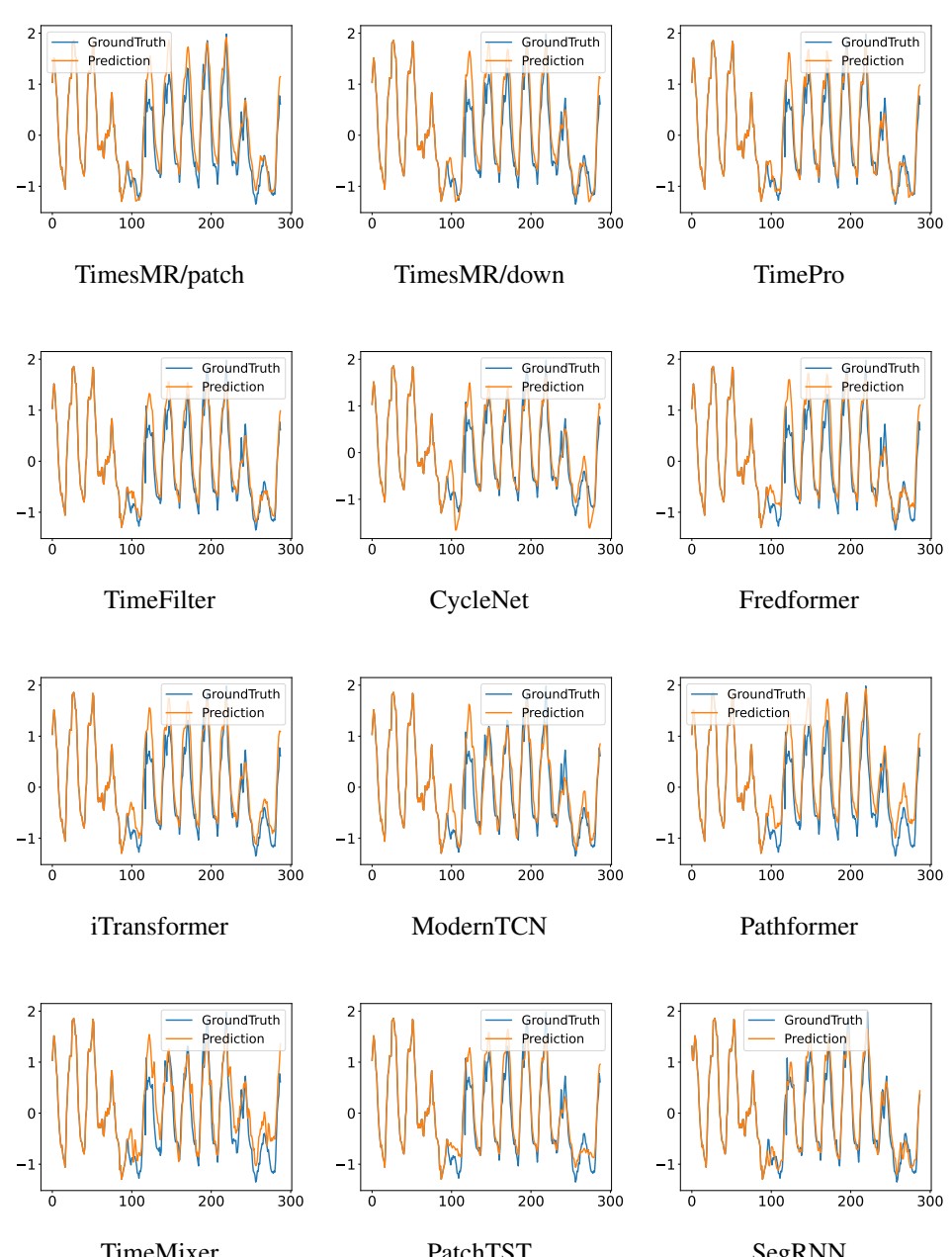

Figure 13: Visualization of input-96-predict-192 results on the ECL dataset.

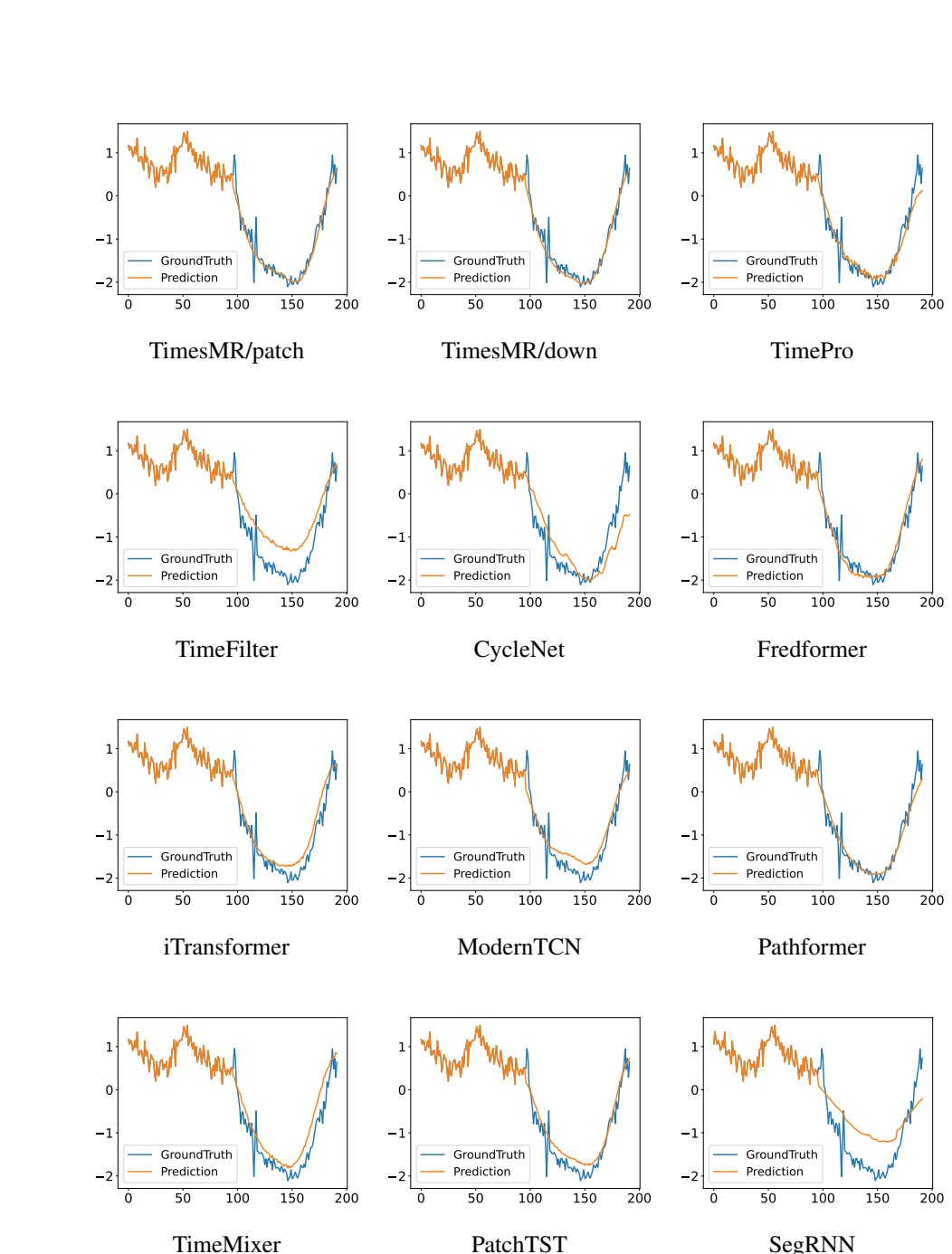

Figure 14: Visualization of input-96-predict-96 results on the PEMS07 dataset.

