# OpenReview forum: "TimesMR: Unlocking the Potential of MLP and RNNs for Multivariate Time Series Forecasting"
_ICLR.cc/2026/Conference — Submitted to ICLR 2026_

### Official Review · Reviewer_vvAx · 2025-10-28

**Soundness:** 3
**Presentation:** 3
**Contribution:** 2
**Rating:** 4
**Confidence:** 4

**Summary:**

The authors address two core challenges in MTSF: the high complexity of existing temporal models and the computational cost and noise-sensitivity of attention-based variable modeling in high-dimensional data.

To solve this, the authors propose **TimesMR**, an "effective and simple" architecture composed of two core modules:
1.  **Multi-scale MLP Module**: This module replaces complex temporal feature extractors with two lightweight variants (Multi-patch MLP and Multi-downsampling MLP) to efficiently capture multi-scale temporal patterns.
2.  **Grouped Bidirectional RNNs (Grouped BiRNNs)**: This module is proposed for efficient variable correlation modeling. The authors argue that the gating mechanism of RNNs (specifically GRU) helps suppress noise. To manage high dimensionality, variables are divided into $G$ groups (e.g., $G=\sqrt{C}$), processing "intra-group" correlations in parallel before an "inter-group" BiRNN exchanges information, reducing complexity to $\mathcal{O}(\sqrt{C})$.

The authors conduct comprehensive experiments on 16 datasets against 18 baselines, demonstrating that TimesMR achieves state-of-the-art (SOTA) or competitive performance in the vast majority of cases.

**Strengths:**

* **Strong Empirical Performance**:
    The most significant contribution of this paper is its SOTA results across an exceptionally broad set of benchmarks. TimesMR consistently outperforms 18 strong baselines (including iTransformer, Pathformer, and TimeMixer) on 16 datasets, making this a solid and convincing empirical contribution.

* **Pragmatic and Efficient Architectural Design**:
    The authors correctly identify the computational pain points of current SOTA models (like Transformers). The use of lightweight MLPs and RNNs as replacements is a very pragmatic design choice.
    The Grouped BiRNN module strikes a good balance between efficiency and performance. The analysis shows its computational complexity is $\mathcal{O}(\sqrt{C} \cdot D^2)$, which is superior to attention's $\mathcal{O}(C^2 \cdot D)$. Experiments confirm that TimesMR has a significantly faster training time on high-dimensional datasets (e.g., Traffic, PEMS07) than models like iTransformer.

* **Strong Component-wise Analysis**:
    The experimental design is thorough. Beyond SOTA comparisons, the paper validates the "plug-and-play" capability of its modules. Furthermore, it provides a direct comparison of Grouped BiRNN against Grouped Attention and a cost-benefit analysis for different group numbers, all of which greatly strengthen the paper's claims.

**Weaknesses:**

* **Architectural Redundancy**:
    The paper proposes two parallel Multi-scale MLP modules: Multi-patch and Multi-downsampling. However, the experimental results show their performance is nearly identical. Critically, the authors confirm in Appendix C.1 that combining them yields no performance benefit, attributing this to "inherent overlap" and "redundancy". This raises the question of whether it is reasonable to propose two highly redundant modules. The paper does not seem to describe any interaction between them (they are presented as an either/or choice in Fig. 3b), which makes the final model design feel less concise than it could be.

* **Incomplete Discussion of Grouping Strategy**:
    The paper's discussion of the Grouped BiRNN strategy is incomplete, focusing on efficiency while omitting key methodological details and trade-offs.
    * **Missing Methodology (The "How")**: The paper **completely omits *how* variables are assigned to groups**. Are they grouped sequentially (as they appear in the dataset), randomly, or based on some pre-calculated correlation? This is a critical missing detail. The formation of these local groups directly dictates which variables can interact within the "intra-group" BiRNN, and a poor grouping choice (e.g., splitting two highly correlated variables) could significantly hamper performance.
    * **Motivation (The "Why")**: The motivation appears to be **efficiency, not performance**. From an information-theoretic perspective, not grouping ($G=1$) is "lossless," while grouping ($G>1$) is a "lossy" approximation. The experimental results in Appendix B.4 seem to confirm this: **no grouping ($G=1$)** actually achieves the **best (lowest) MSE** on multiple datasets. This strongly suggests the strategy is an engineering trade-off that sacrifices accuracy for speed.

* **Unanalyzed Padding and Truncation Flow**:
    When the number of variables $C$ is not divisible by the group number $G$, the model must use Padding up to $C'$. This padded variable then participates in the "intra-group BiRNN" computation. At the end of the process, the model "truncates" the output, discarding the padded variable's representation. The paper **does not specify what padding method is used** and **provides no analysis** of whether this "compute-then-discard" process has an unintended impact on the vector representations of the *other real variables* within the same group.


* **Unclear Motivation for BiRNN Integration**:
	The introduction of the BiRNN component lacks a clear and compelling design motivation, which raises concerns that the overall contribution may be perceived as a **progressive extension combining PatchTST and BiRNN** rather than a genuinely novel architectural advancement. The paper does not articulate: * **What** specific temporal dependencies BiRNN captures that patch-based MLPs or Transformers cannot; * **Why** this integration is essential instead of being an optional enhancement; * **How** BiRNN interacts with or complements the multi-patch representation in a principled way. Without addressing these questions, readers may question whether the incorporation of BiRNN is fundamentally necessary, potentially weakening the perceived scientific significance and originality of the proposed model.

**Questions:**

1. Given that the Multi-patch and Multi-downsampling modules are demonstrated to be functionally redundant, could the authors discuss the necessity of retaining both? Would one module suffice? Or, is there any latent interaction between these two modules that was not discussed?

2.  I have two questions about the group strategy：
    * Could the authors first clarify **what method** is used to assign variables to different groups? (e.g., sequential, random, correlation-based?) This is a critical, un-discussed detail.
    * Following from Figure 5, is it correct to conclude that grouping ($G>1$) is **purely** a trade-off for computational efficiency, and does not in itself offer a performance benefit over the "lossless" $G=1$ case?

3. I have two questions about the "pad-truncate" flow:
    * Could the authors first clarify what specific padding method (e.g., zero-padding, mean-padding, etc.) is used when the variable dimension $C$ is padded to $C'$?
    * Could the authors provide an analysis of the specific impact this "compute-then-discard" flow has on model accuracy? Does the padded variable negatively affect the representations of the real variables within its group during the intra-group computation?

**Details Of Ethics Concerns:**

1.  The paper proposes two functionally redundant Multi-scale MLP modules (Multi-patch and Multi-downsampling), which adds unnecessary architectural complexity.
2.  The Grouped BiRNNs strategy appears to be an engineering trade-off that sacrifices minor model accuracy for computational speed, rather than a performance-enhancing mechanism. The paper also fails to specify *how* variables are grouped.
3.  The paper fails to specify the padding method used and does not analyze the potential impact of the "pad-truncate" flow on model representations.

---

> ### Author Response · Authors · 2025-11-25
>
> We appreciate your recognition of our strengths. We have addressed all your comments below.
>
> #### **W1&Q1. Architectural Redundancy: The paper proposes two parallel Multi-scale MLP modules: Multi-patch and Multi-downsampling...This raises the question of whether it is reasonable to propose two highly redundant modules. The paper does not seem to describe any interaction between them.  Given that the Multi-patch and Multi-downsampling modules are demonstrated to be functionally redundant, could the authors discuss the necessity of retaining both? Would one module suffice? Or, is there any latent interaction between these two modules that was not discussed?**
>
> **Response:** We clarify that the Multi-patch and Multi-downsampling techniques are designed as two alternative approaches within our Multi-scale MLP module, rather than being used simultaneously. As described in **Section 5.1 (lines 334–335)**, in TimesMR we only use one module at a time: TimesMR/patch employs Multi-patch, while TimesMR/down employs downsampling, and their performances are reported separately.
>
> The rationale for providing both Multi-patch and Multi-downsampling modules is to enhance the versatility of our method across diverse datasets. As analyzed in **Appendix C.1**, we present a direct comparison of the patch and downsampling variants in our Multi-scale Module. Specifically, we visualize fragments of datasets with varying characteristics and report the performance of both variants in **Figure 9**. The results demonstrate that the patch variant is effective for capturing local temporal variations by segmenting the input sequence into non-overlapping patches and extracting intra-patch features via MLPs. This allows the model to focus on fine-grained, high-frequency patterns such as local fluctuations or short-term changes (e.g., ETTm1 and Traffic datasets). In contrast, the downsampling variant is better suited for modeling global or long-term temporal patterns by reducing temporal resolution, aggregating information over larger time spans, and emphasizing overall trends and long-term changes (e.g., ETTh2 and METR-LA datasets).

---

> > ### Author Response · Authors · 2025-11-25
> >
> > #### **W2&Q2. More discussion about Grouping Strategy: How variables are assigned to groups; the motivation of grouping.**
> >
> > **Response:** As suggested, we have clarify how variables are assigned to groups and further elaborate on the motivation behind our grouping strategy.
> >
> > **How variables are assigned to groups?**
> > We clarify that we do not utilize any prior ordering or domain knowledge when forming groups; instead, we follow the input order of the variables as presented in the dataset. Thus, variables are grouped sequentially according to their original dataset order. In multivariate time series forecasting, variable order can be arbitrary in real-world scenarios. Our approach is intentionally designed to avoid dependence on any specific variable ordering or prior knowledge of variable relationships, ensuring general applicability and robustness.
> >
> > Suppose there are $C$ variables and we want to divide them into $G$ groups with $g$ variables per group. We first pad the variable set from $C$ to $C'$ (padding strategy is analyzed below), so that $C' = G \times g$. Then, the first $g$ variables in $C'$ form the first group, the next $g$ variables form the second group, and so on until all groups are assigned.
> >
> > To further assess whether the grouping strategy affects performance, we conducted new experiments to compare alternative grouping variants: (1) Random grouping, where the variable order is randomly permuted before grouping, so each group contains an arbitrary set of variables. (2) Correlation-based grouping, where variables are clustered according to their pairwise correlations, with the number of clusters set equal to the number of groups, so highly correlated variables are placed within the same group. We report the results in **Table 1 below**. We observe that both the random and correlation-based grouping strategies result in only minimal changes to our model's performance. This demonstrates that our method robustly captures dependencies among variables through intra-group and inter-group modeling, and is not sensitive to the specific grouping scheme. The model's ability to learn inter-variable relationships does not depend on any particular grouping configuration.
> >
> >
> >
> > **Motivation of Grouping Strategy.**
> > As described in **lines 274–278 and 952–956 of the revised paper**, the primary motivation for our grouping strategy is to reduce the training cost of the BiRNN while maintaining comparable performance. We conducted extensive experiments to analyze this effect, as shown in Appendix B.4 and **Figure 7 in the revised paper**. When varying the group number, the time cost is significantly reduced for group number $G>1$ compared to $G=1$ (no grouping), while the effectiveness remains similar. Specifcially, when $G=5$, on datasets PEMS04, PEMS07, ECL, and Traffic, the effectiveness performance (MSE) is slightly better or the same as with $G=1$. Given that the performance remains comparable, we believe the efficiency improvement in training time justifies the use of the grouping strategy, which is the main motivation for this design of grouped BiRNN.
> >
> >
> > **Table 1. Comparison of the different grouping methods**
> > |Model|metric|Traffic|ECL|PEMS-BAY|METR-LA|PEMS03|PEMS04|PEMS07|PEMS08
> > |---|---|---|---|---|---|---|---|---|---|
> > |TimesMR/patch|MSE|0.426|0.161|0.365|0.735|0.097|0.084|0.073|0.108|
> > ||MAE|0.280|0.255|0.273|0.460|0.197|0.186|0.171|0.198|
> > |random|MSE|0.428|0.160|0.364|0.732|0.092|0.083|0.072|0.108|
> > ||MAE|0.282|0.255|0.275|0.470|0.196|0.186|0.169|0.201|
> > |correlation-based|MSE|0.431|0.164|0.368|0.734|0.094|0.083|0.072|0.112|
> > ||MAE|0.280|0.259|0.274|0.458|0.198|0.185|0.167|0.199|

---

> > > ### Author Response · Authors · 2025-11-25
> > >
> > > #### **W3&Q3. Specify what padding method is used and provides no analysis of whether this "compute-then-discard" process has an unintended impact on the vector representations of the other real variables within the same group.**
> > >
> > > **Response:** As suggested, we have provided the detailed explanation and experiments of the pad and truncate flow below.
> > >
> > > First, for padding, if the number of variables $C$ is not divisible by the group number $G$, we pad the missing variables at the end by repeating the first few variables to obtain $C'$ variables, where $C' = G \cdot g$, ensuring that all groups have equal size. The $C'$ variables are then sequentially assigned to the $G$ groups. After the intra-group and inter-group BiRNN computations, we truncate the output by discarding the representations corresponding to the padded variables.
> > >
> > > Second, to assess the impact of this "compute-then-discard" process, we experimentally compare several padding strategies: zero-padding (pad with zeros), mean-padding (pad with the mean value of the $C$ variables), and latter-padding (pad by repeating the last few variables). The results in **Table 2 below** show that the choice of padding strategy has only a minor effect on model performance, with most variations within 1\%. This demonstrates that the padded variables do not adversely affect the representations of the real variables, and our model is robust to different padding approaches.
> > >
> > > **Table 2 The impact of different padding strategies**
> > > |Model|metric|Traffic|ECL|PEMS-BAY|METR-LA|PEMS03|PEMS04|PEMS07|PEMS08
> > > |---|---|---|---|---|---|---|---|---|---|
> > > |TimesMR/patch|MSE|0.426|0.161|0.365|0.735|0.097|0.084|0.073|0.108|
> > > ||MAE|0.280|0.255|0.273|0.460|0.197|0.186|0.171|0.198|
> > > |zero padding|MSE|0.428|0.162|0.368|0.725|0.097|0.084|0.073|0.105|
> > > ||MAE|0.281|0.256|0.272|0.467|0.197|0.187|0.171|0.196|
> > > |mean padding|MSE|0.427|0.162|0.368|0.730|0.098|0.084|0.073|0.107|
> > > ||MAE|0.280|0.257|0.273|0.473|0.197|0.187|0.171|0.196|
> > > |latter padding|MSE|0.424|0.163|0.366|0.722|0.097|0.084|0.074|0.106|
> > > ||MAE|0.279|0.258|0.272|0.470|0.197|0.187|0.171|0.196|

---

> > > > ### Author Response · Authors · 2025-11-25
> > > >
> > > > #### **W4. Motivation for BiRNN Integration:  What specific temporal dependencies BiRNN captures that patch-based MLPs or Transformers cannot;  Why this integration is essential instead of being an optional enhancement;  How BiRNN interacts with or complements the multi-patch representation in a principled way.**
> > > >
> > > >
> > > > **Response:** In what follows, we first explain why and what is captured by the BiRNN, and then explain how BiRNN complements the multi-patch representation in a principled way.
> > > >
> > > > *What specific temporal dependencies BiRNN captures that patch-based MLPs or Transformers cannot.*
> > > > *Why this integration is essential instead of being an optional enhancement.*
> > > >
> > > > Firstly, we clarify that patch-based methods such as PatchTST combine patching with attention mechanisms to independently model temporal dependencies within each variable, but do not address relationships across variables. In contrast, our Grouped BiRNN is specifically designed to efficiently capture complex inter-variable relationships rather than temporal dependencies. This is crucial because, in multivariate time series, some variables are strongly correlated while others are weakly related or even unrelated. Achieving robust performance requires a mechanism that can emphasize informative correlations and suppress the influence of weak or irrelevant variables. RNN architectures such as LSTMs and GRUs are particularly effective for this purpose due to their gating mechanisms, which filter out unrelated interference and selectively retain relevant information. This motivates our exploration of the underutilized potential of RNNs for modeling variable relationships.
> > > >
> > > > The BiRNN in our paper is thus focused on modeling variable relationships, not temporal dependencies. While MLPs and Transformer-based attention mechanisms can also be used for inter-variable modeling, each has limitations. Attention, as shown in **Figure 2**, can capture strong correlations but often degrades performance when correlations are weak or irrelevant, since it aggregates information indiscriminately from all variables. RNNs, by contrast, suppress interference from weakly related variables and selectively retain useful information, leading to improved performance on datasets with weak inter-variable correlations (e.g., ETTm1, ETTm2, Weather), as shown in **Table 2 and Table 3 in the revised paper**. MLPs, although lightweight, lack the capacity to selectively emphasize informative variables and suppress unrelated or weakly relevant ones. Our BiRNN component naturally aggregates useful information across variables and avoids interference from weakly related variables, resulting in better variable representations. As shown in **Figure 11 in the revised paper**, our method consistently outperforms MLP-based approaches across diverse scenarios with varying variable relationships.
> > > >
> > > > *How BiRNN interacts with or complements the multi-patch representation in a principled way.*
> > > >
> > > > In our design, the multi-scale MLP module and the grouped BiRNN are designed to capture complementary aspects of multivariate time series data. The multi-scale MLP module is responsible for modeling temporal dependencies within each variable, while the grouped BiRNN is subsequently employed to capture inter-variable relationships. This principled two-stage approach enables effective disentanglement of temporal and variable relationship modeling, leveraging the strengths of each component, as illustrated in Figure 3. Specifically, for a multivariate time series $\mathcal X_{t-L+1:t} \in \mathbb{R}^{C \times L}$, we first input it into the MLP-based multi-scale module, which utilizes either the multi-patch MLP or the multi-downsampling MLP to extract temporal dependencies. After this step, the series is embedded along the temporal dimension $L$ into a hidden representation for each variable, yielding $\mathcal{X_tem} \in \mathbb{R}^{C \times D}$, where $D$ denotes the hidden dimension. Next, $\mathcal{X_tem}$ is processed through $N$ Variable Correlation Module layers with grouped BiRNNs to model complex inter-variable relationships, resulting in enhanced variable representations $\mathcal{X_var} \in \mathbb{R}^{C \times D}$. Finally, a linear transformation is applied to $\mathcal{X_var}$ along the hidden dimension to produce the final prediction $\hat{\mathcal{X}}_{t+1:t+H} \in \mathbb{R}^{C \times H}$, where $H$ is the forecasting horizon.

---

### Official Review · Reviewer_H2Md · 2025-10-31

**Soundness:** 3
**Presentation:** 3
**Contribution:** 2
**Rating:** 4
**Confidence:** 3

**Summary:**

The paper proposes TimesMR, a multivariate time-series forecasting model that combines (i) two lightweight multi-scale MLP temporal modules and (ii) a Grouped BiRNN variable-correlation module that processes variables to exchange intra- and inter-group information efficiently. Across 16 datasets and against 18 baselines, the authors report state-of-the-art average MSE/MAE, plus ablations showing each module’s contribution and efficiency analyses versus input length. Code is given as an anonymous link.

**Strengths:**

1. The proposed method achieves broad, consistent gains on 16 datasets, including high-dimensional traffic/PEMS benchmarks. Ablations show each module matters. Efficiency curves vs. input length are informative.
2. Architecture and data flow (patching/downsampling, RevIN, Grouped BiRNN intra/inter passes, fusion) are well illustrated.
3. Results indicate a compelling accuracy/efficiency trade-off for high-dimension settings where attention can be costly or noisy.

**Weaknesses:**

1. Temporal modules are close to prior multi-scale designs but implemented with MLPs; the variable module revisits RNNs with grouping. The paper’s contribution is engineering-oriented rather than theoretically or algorithmically deep.
2. Using dataset-level mean Pearson correlation (MCC) as a noise proxy is limited; correlation structure is often non-linear, time-varying, or lagged. Stronger diagnostics (e.g., partial correlations, Granger-style tests, lagged cross-correlations, spectral coherence) or synthetic studies would be helpful.
3. Given rising strong non-attention baselines and foundation-model-style forecasters, a brief comparison or discussion would be valuable even if only to argue scope differences.

**Questions:**

1. For Fig. 1, how was lookback length “searched”? Was the same protocol (search space, budget, early stopping) applied to all baselines? Please clarify the validation split and ensure no test leakage.
2. Could you report per-dataset standard deviations over multiple seeds and perform paired significance tests (e.g., Wilcoxon) against the strongest baseline(s)?

---

> ### Author Response · Authors · 2025-11-25
>
> We appreciate your recognition of our strengths. We have addressed all your comments below.
>
> #### **W1. Temporal modules are close to prior multi-scale designs but implemented with MLPs; the variable module revisits RNNs with grouping. The paper's contribution is engineering-oriented rather than theoretically or algorithmically deep.**
>
> **Response:** We would like to clarify that the primary goal of our work is to design a simple yet effective and efficient architecture for multivariate time series forecasting, in contrast to existing complex models. Achieving simplicity is often more challenging, as it requires thoughtful **technical designs** of the proposed modules and the overall architecture, as shown in **Figure 3 of the revised paper**.
>
> Specifically, our MLP-based multi-scale module (Section 4.1) is designed to efficiently capture temporal features. While prior local time series embedding methods have demonstrated strong performance, they typically rely on heavy components such as attention mechanisms, which incur high computational costs. In contrast, our proposed multi-scale MLP module, when combined with multi-scale techniques, demonstrates that a lightweight MLP is capable of capturing rich temporal patterns. This is empirically supported by observations in **Figure 1**, where our approach achieves competitive or superior performance with significantly reduced time and memory requirements. To capture temporal dependencies, we combine multi-scale techniques with MLPs, showing that even simple MLPs can achieve competitive or superior performance in modeling temporal patterns (**as shown in Figure 1, Table 11 and Table 12 in the revised paper**).
>
> Furthermore, as demonstrated in Figure 3 of the revised paper, the architecture of TimesMR is carefully designed by integrating the MLP-based multi-scale module with the proposed variable correlation module. The variable correlation module (Section 4.2) introduces a novel grouped bidirectional RNN (Grouped BiRNN) approach for modeling variable correlations. Bidirectional RNNs aggregate relevant information from other variables and suppress noise from weakly correlated ones. This design leads to improved performance and computational efficiency (**as shown in Figure 4, Figure 11 and Figure 12 in the revised paper**). For high-dimensional data, variables are grouped, and intra-group correlations are captured by applying bidirectional RNNs within each group. Inter-group information is exchanged via the final hidden states of each group's RNN, significantly reducing memory and computation costs while preserving effective correlation modeling. Unlike attention, which incurs quadratic complexity when modeling relationships among variables, our grouped BiRNN method has a computational complexity of $\mathcal{O}(\sqrt{C} \cdot D^{2})$ (see **Appendix D**), scaling with the square root of the number of variables $C$.
>
>
> Overall, the results in Section 5.2 demonstrate the superior performance of our TimesMR model compared to other baselines, including the multi-scale MLP module and the variable correlation module. In Section 5.3 and the Appendix, we provide extensive experimental analysis to demonstrate the efficiency of the proposed method and validate the effectiveness of each component in our model.

---

> > ### Author Response · Authors · 2025-11-25
> >
> > #### **W2. Using dataset-level mean Pearson correlation (MCC) as a noise proxy is limited; correlation structure is often non-linear, time-varying, or lagged. Stronger diagnostics (e.g., partial correlations, Granger-style tests, lagged cross-correlations, spectral coherence) or synthetic studies would be helpful.**
> >
> > **Response:** As suggested, in addition to the dataset-level mean Pearson correlation (MCC) analysis presented, we further analyze the noise characteristics in multivariate time series datasets using partial correlations and spectral coherence, with visualization results reported in **Figure 5 and Figure 6 in the revised paper** to provide stronger diagnostics.
> >
> > Specifically, Partial correlation measures the direct linear relationship between two variables while controlling for the influence of all other variables. This allows us to better identify the true pairwise dependencies that are not confounded by other correlated variables. When the number of variables is large, the distribution of partial correlations tends to be biased toward lower values, as controlling for many other variables often reduces the residual linear dependency between a pair of variables. Therefore, in our implementation, for each variable, we select the top-10 most strongly correlated variables as control variables rather than using all other variables. This strategy focuses on the most relevant confounding factors, reduces computational cost in high-dimensional settings, and avoids diluting the residual signal with weakly correlated variables. As a result, the partial correlations accurately reflect the meaningful direct dependencies between variable pairs. Spectral coherence quantifies the degree of linear correlation between two time series at different frequencies. It helps capture frequency-dependent dependencies and lagged relationships, providing a complementary view of the interactions that cannot be observed with simple time domain correlations.
> >
> > From **Figure 5 and Figure 6**, we observe that the partial correlations in most datasets are centered around zero, indicating that the direct linear dependencies between variable pairs are generally weak. Similarly, the spectral coherence for most datasets is also close to zero, suggesting that frequency-dependent relationships and lagged correlations are limited. This implies that these datasets contain a relatively high level of noise, which requires the model to accurately capture the meaningful variable relationships while avoiding the influence of weakly related or unrelated variables.
> >
> > The new findings from partial correlations and spectral coherence, together with the original MCC analysis, further validate our motivation for designing the Grouped BiRNN module to aggregate useful information from other variables and suppress the influence of variable noise from weakly correlated variables.

---

> > > ### Author Response · Authors · 2025-11-25
> > >
> > > #### **W3. Given rising strong non-attention baselines and foundation-model-style forecasters, a brief comparison or discussion would be valuable even if only to argue scope differences.**
> > >
> > > **Response:** As suggested, we have revised the related work to include both non-attention baselines and foundation-model-style forecasters. Specifically, our discussion addresses two key aspects: temporal dependency modeling and variable relationship modeling.
> > >
> > > Modeling temporal dependency is central to time series forecasting. In recent years, deep learning models have shown great promise and can be broadly categorized into four paradigms: Transformer-based, CNN-based, RNN-based, and MLP-based models. Transformer-based models leverage attention mechanisms to capture dependencies between tokens and have proven effective in time series analysis. However, the quadratic complexity of attention limits scalability, prompting optimizations such as Informer, Autoformer, FEDformer, and PatchTST. CNN-based models, including ModernTCN, TimesNet, and MICN, use convolutional kernels to capture local temporal patterns but often require deep stacking to achieve global modeling, which can affect training stability. RNN-based models, such as SegRNN and LSTNet, utilize recurrent structures to model temporal dependencies but may encounter vanishing or exploding gradients. MLP-based models have recently gained attention for their lightweight frameworks, with models like DLinear and TSMixer demonstrating competitive performance.
> > > Recently, time series foundation models have emerged, particularly for zero-shot and few-shot scenarios. These models typically follow two approaches: pretraining on large-scale time series datasets to learn diverse temporal patterns (e.g., Chronos, TimesFM, MOMENT), or leveraging prompt embeddings and alignment with language models to enhance temporal representations (e.g., Time-LLM, CALF). Foundation models represent a shift toward large-scale, pretrainable architectures for time series data, facilitating knowledge transfer across datasets. As these models are beyond the scope of our work, we do not include them as baselines in our experimental comparisons. Overall, multi-scale modeling has become mainstream for capturing complex temporal patterns, as demonstrated by methods such as TimeMixer, Pathformer, TimesNet, and MICN. Notably, TimeMixer and Pathformer are recent multi-scale methods, but their embedding layers, mixing architectures, and attention mechanisms introduce substantial time and memory overhead. To address this, we propose two novel multi-scale MLP modules that are both efficient and effective in capturing temporal dependencies, offering a simpler yet powerful alternative. We also compare the performance of non-attention baselines in Table 2 and Table 3 of the revised paper.
> > >
> > > Variable relationship modeling in multivariate time series forecasting has received significant attention, with existing methods categorized as Variable Dependent, Variable Independent, Attention-based, Cluster-based, MLP-based, CNN-based, Mamba-based, and Graph-based approaches. Variable Dependent methods embed all variables into a hidden vector and focus on temporal dependencies, increasing representational capacity but often lacking explicit modeling of variable relationships and generalization (e.g., Informer, Autoformer, FEDformer, MICN, TimesNet, Pathformer, TimeMixer). Variable Independent methods treat each variable as independent, improving generalization but limiting representation and ignoring complex inter-variable relationships (e.g., PatchTST, SparseTSF, CycleNet). Attention-based methods model variable relationships via attention (e.g., TQNet, iTransformer, Leddam, CARD, Crossformer, Fredformer). Cluster-based methods group correlated variables (e.g., DUET). MLP-based methods use MLPs to model variable correlations (e.g., TSMixer, SOFTS). CNN-based methods employ convolutional architectures (e.g., ModernTCN). Mamba-based methods adapt Mamba for inter-variate correlation extraction (e.g., TimePro, SMamba). Graph-based methods use graph structures to model variable correlations, with nodes representing variables and edges representing relationships (e.g., TimeFilter). In summary, attention-based methods excel at capturing token dependencies but incur high computational costs for high-dimensional datasets, while MLP-based methods are efficient but may struggle with complex variable relationships. In this work, we revisit the potential of RNNs for modeling variable relationships and propose grouped bidirectional RNNs. This approach divides variables into groups, enabling efficient intra-group and inter-group correlation modeling, thereby capturing complex relationships with reduced computational overhead. Extensive experiments validate the effectiveness and efficiency of our techniques.

---

> > > > ### Author Response · Authors · 2025-11-25
> > > >
> > > > #### **Q1. For Fig. 1, how was lookback length "searched"? Was the same protocol (search space, budget, early stopping) applied to all baselines? Please clarify the validation split and ensure no test leakage.**
> > > > **Response:** In Fig. 1, "searched lookback length" means that we selected the lookback length from the set {96, 192, 336, 576, 720}, choosing the value that yielded the best forecasting performance for each method. This is widely adopted in recent literature for evaluation (e.g., PatchTST, TimeMixer and ModernTCN). To ensure a fair comparison, we applied the search space, stopping criteria, and experimental settings to all baselines and our model. Specifically, all models were trained using L2Loss and the ADAM optimizer, with a batch size of 32, for 15 epochs, and early stopping was triggered if the validation loss did not improve for three consecutive epochs.
> > > >
> > > > For the validation split, as described in **Appendix B.1**, we followed the same data preprocessing and train–validation–test split protocol as iTransformer, with detailed split sizes for each dataset reported in Table 7. To prevent test leakage, all datasets were split chronologically: a 6:2:2 ratio for the ETT datasets and PEMS03/PEMS04/PEMS07/PEMS08, and a 7:1:2 ratio for all other datasets. For example, the Traffic dataset uses a chronological 7:1:2 split, resulting in training/validation/test sizes of (12185: 1757: 3509). No test leakage occurred in our experiments.

---

> > > > > ### Author Response · Authors · 2025-11-25
> > > > >
> > > > > #### **Q2. Could you report per-dataset standard deviations over multiple seeds and perform paired significance tests (e.g., Wilcoxon) against the strongest baseline(s)?**
> > > > >
> > > > > **Response:** As suggested, we provide additional results on per-dataset standard deviations and significance tests against the strongest baselines.
> > > > >
> > > > >
> > > > > In **Appendix B.3 Table 9 and Table 10**, we have reported the standard deviations on eight datasets to verify the stability of our methods. Here, we further provide the standard deviations for the remaining seven datasets in **Table 1 below**. Specifically, we use five different random seeds: 2025, 2026, 2027, 2028, and 2029, and report the standard deviation for each forecasting horizon as well as the average across each dataset. The results demonstrate that our model's performance remains stable across different random seeds, confirming the robustness of our method.
> > > > >
> > > > > **Table 1. Per-dataset standard deviations over multiple seeds**
> > > > > |Datasets|metric|TimesMR/patch| | | |  |
> > > > > |---|---|---|---|---|---| ---|
> > > > > | | |96|192|336|720|average
> > > > > |PEMS03|MSE|0.060 $\pm$ 0.000|0.074 $\pm$ 0.000|0.105 $\pm$ 0.002|0.141 $\pm$ 0.003|0.094 $\pm$ 0.001||||
> > > > > ||MAE|0.160 $\pm$ 0.000|0.179 $\pm$ 0.000|0.208 $\pm$ 0.000|0.244 $\pm$ 0.001|0.198 $\pm$ 0.000||||
> > > > > |PEMS04|MSE|0.068 $\pm$ 0.000|0.076 $\pm$ 0.000|0.089 $\pm$ 0.000|0.106 $\pm$ 0.000|0.085 $\pm$ 0.000||||
> > > > > ||MAE|0.168 $\pm$ 0.000|0.178 $\pm$ 0.000|0.194 $\pm$ 0.000|0.213 $\pm$ 0.000|0.188 $\pm$ 0.000||||
> > > > > |PEMS07|MSE|0.053 $\pm$ 0.000|0.063 $\pm$ 0.000|0.077 $\pm$ 0.000|0.096 $\pm$ 0.000|0.072 $\pm$ 0.000|||||
> > > > > ||MAE|0.148 $\pm$ 0.000|0.161 $\pm$ 0.000|0.178 $\pm$ 0.000|0.196 $\pm$ 0.000|0.171 $\pm$ 0.000||||
> > > > > |PEMS08|MSE|0.070 $\pm$ 0.000|0.088 $\pm$ 0.001|0.113 $\pm$ 0.002|0.156 $\pm$ 0.004|0.107 $\pm$ 0.001||||
> > > > > ||MAE|0.169 $\pm$ 0.000|0.184 $\pm$ 0.000|0.204 $\pm$ 0.001|0.231 $\pm$ 0.005|0.197 $\pm$ 0.001||||
> > > > > |METR-LA|MSE|0.433 $\pm$ 0.007|0.612 $\pm$ 0.005|0.819 $\pm$ 0.003|1.037 $\pm$ 0.002|0.726 $\pm$ 0.002||||
> > > > > ||MAE|0.336 $\pm$ 0.002|0.427 $\pm$ 0.006|0.535 $\pm$ 0.006|0.633 $\pm$ 0.003|0.482 $\pm$ 0.003||||
> > > > > |PEMS-BAY|MSE|0.254 $\pm$ 0.001|0.331 $\pm$ 0.000|0.405 $\pm$ 0.001|0.467 $\pm$ 0.002|0.364 $\pm$ 0.001||||
> > > > > ||MAE|0.225 $\pm$ 0.000|0.256 $\pm$ 0.000|0.290 $\pm$ 0.000|0.321 $\pm$ 0.001|0.273 $\pm$ 0.000||||
> > > > > |PEMSD7|MSE|0.239 $\pm$ 0.001|0.308 $\pm$ 0.001|0.364 $\pm$ 0.002|0.397 $\pm$ 0.002|0.327 $\pm$ 0.000||||
> > > > > ||MAE|0.260 $\pm$ 0.003|0.301 $\pm$ 0.001|0.332 $\pm$ 0.003|0.357 $\pm$ 0.002|0.312 $\pm$ 0.001||||
> > > > >
> > > > > Moreover, we perform paired significance tests against the strongest baselines, including TimeFilter and TimePro, under five different random seeds using MSE as the evaluation metric (lower values indicate better performance). The Wilcoxon signed–rank test is not applicable in this case because our method outperforms these baselines on all datasets, and Wilcoxon requires both positive and negative paired differences. Therefore, we use the one-sided Sign Test, which considers only the direction of paired differences and remains valid when all differences share the same sign. The results (p-value = 0.03125), summarized in **Table 2 below**, demonstrate that our method significantly outperforms the strongest baselines across all datasets. This provides strong statistical evidence that the observed improvements are consistent and not due to random variation, but instead reflect genuine performance gains.
> > > > >
> > > > > **Table 2. Sign Test against TimeFilter and TimePro**
> > > > > |Datasets|Model|different seeds| | | |  | p-value of Sign Test
> > > > > |---|---|---|---|---|---| --|---|
> > > > > | ||2025|2026|2027|2028|2029|
> > > > > |PEMS03|TimesMR/patch|**0.095**|**0.092**|**0.095**|**0.095**|**0.096**|
> > > > > ||TimeFilter|0.120|0.121|0.121|0.121|0.121|0.03125
> > > > > ||TimePro|0.116|0.117|0.115|0.117|0.120|0.03125
> > > > > |PEMS04|TimesMR/patch|**0.085**|**0.085**|**0.085**|**0.085**|**0.084**|
> > > > > ||TimeFilter|0.120|0.119|0.121|0.122|0.122|0.03125
> > > > > ||TimePro|0.117|0.118|0.122|0.117|0.117|0.03125
> > > > > |PEMS07|TimesMR/patch|**0.097**|**0.097**|**0.096**|**0.097**|**0.096**|
> > > > > ||TimeFilter|0.100|0.104|0.099|0.102|0.101|0.03125
> > > > > ||TimePro|0.102|0.104|0.102|0.103|0.102|0.03125
> > > > > |PEMS08|TimesMR/patch|**0.107**|**0.107**|**0.109**|**0.104**|**0.107**|
> > > > > ||TimeFilter|0.152|0.155|0.147|0.146|0.147|0.03125
> > > > > ||TimePro|0.252|0.264|0.247|0.249|0.262|0.03125
> > > > > |METR-LA|TimesMR/patch|**0.726**|**0.728**|**0.729**|**0.726**|**0.722**|
> > > > > ||TimeFilter|0.821|0.819|0.825|0.821|0.821|0.03125
> > > > > ||TimePro|0.746|0.753|0.748|0.757|0.753|0.03125
> > > > > |PEMS-BAY|TimesMR/patch|**0.364**|**0.363**|**0.366**|**0.363**|**0.366**|
> > > > > ||TimeFilter|0.555|0.589|0.572|0.550|0.559|0.03125
> > > > > ||TimePro|0.451|0.448|0.453|0.447|0.449|0.03125
> > > > > |PEMSD7|TimesMR/patch|**0.326**|**0.327**|**0.326**|**0.327**|**0.327**|
> > > > > ||TimeFilter|0.510|0.506|0.515|0.519|0.516|0.03125
> > > > > ||TimePro|0.381|0.383|0.384|0.384|0.384|0.03125

---

> ### Comment · Reviewer_H2Md · 2025-11-27
>
> I appreciate the response and the new experiments, which have addressed my concerns regarding the technical aspects. I understand that the main goal is to reduce computational overhead while maintaining strong forecasting accuracy, and the authors have demonstrated solid empirical results. Nevertheless, I still believe the paper lacks a strong justification for the architectural design, especially after reading the authors’ response to Reviewer vvAx. What, then, are the main research (or theoretical) claims the authors intend to make beyond the empirical superiority?

---

> > ### Author Response · Authors · 2025-11-28
> >
> > **Response:** We are glad to know that our response and new experiments have addressed your technical concerns.
> >
> > Regarding our main research claims, our overarching goal is to develop a simple yet effective architecture, TimesMR, for multivariate time series forecasting. We posit that achieving architectural simplicity while maintaining strong performance itself is already a significant research challenge. Consequently, our contribution focuses on principled research designs and empirical superiority rather than theoretical complexity.
> >
> > Specifically, TimesMR introduces core innovations in two key areas, as illustrated in Figure 3: *(1) temporal dependency modeling within individual variables* and *(2) variable relationship modeling across variables*. We elaborate on these contributions below.
> >
> > **(1) Regarding capturing temporal dependencies:** Recent methods increasingly utilize multi-scale techniques to capture complex patterns like diverse local dynamics. However, many existing architectures rely on sophisticated, computationally expensive components (e.g., attention in Pathformer) for intra- and inter-scale modeling. This raises a natural **research question**: *Are such heavy designs necessary for effective temporal modeling, and can we achieve strong performance with greater efficiency?*
> >
> > To answer this question, we design the **MLP-based multi-scale module** in Section 4.1, which combines lightweight MLPs with multi-scale decomposition to capture rich temporal patterns efficiently. Our results demonstrate that this simple design is sufficient for capturing complex temporal dynamics without resorting to expensive architectural mechanisms. As shown in Figure 1, Table 2, Table 3, Tables 11-13 of the revised paper, our lightweight multi-scale MLP consistently outperforms more complex multi-scale architectures. This validates the core architectural principle of TimesMR: leveraging simple yet effective components to achieve strong performance with low computational cost. Achieving this balance is non-trivial, as naive combinations often yield suboptimal results. By carefully designing TimesMR for synergistic operation, we provide new insight into the design space of time-series models, offering a principled alternative that effectively balances simplicity with predictive accuracy.
> >
> > **(2) Regarding modeling variable correlation:** Recent methods primarily rely on attention mechanisms. However, as shown in Figure 2, while attention excels at capturing strong correlations, it degrades when correlations are weak or irrelevant. Since real-world multivariate time series often mix strong and weak relationships, attention mechanisms can be suboptimal. Furthermore, attention suffers from quadratic complexity with respect to the number of variables, becoming prohibitive in high-dimensional settings, as illustrated in Figure 11.
> >
> > These observations lead to a second **research question**: *How can we effectively and efficiently model complex inter-variable relationships in multivariate time series, particularly when both strong and weak correlations coexist?* Consequently, we identify two *research challenges*: (1) robustly handling complex inter-variable relationships where many variables are only weakly related or unrelated, and (2) reducing computational costs when the number of variables is large.
> >
> > To address these challenges, we propose the **Grouped BiRNN structure** in Section 4.2. By leveraging gating mechanisms, the BiRNN naturally emphasizes informative correlations while suppressing weak or irrelevant ones, making it particularly well-suited for capturing intricate variable relationships. Furthermore, our grouped design significantly reduces computational costs without compromising performance. As shown in Tables 2-4 and Figures 11-12, the Grouped BiRNN achieves superior accuracy with substantial efficiency gains compared to attention-based approaches—for example, reducing inference time from 100s to 42s on PEMS07 and from 118s to 38s on Traffic. These results consistently demonstrate the superiority of our design in both effectiveness and efficiency.
> >
> > Overall, our work makes the following main research contributions:
> > (1) We design a simple MLP-based multi-scale module that effectively captures complex temporal dependencies, challenging the necessity of heavy architectural designs; (2) We propose a grouped BiRNN approach that efficiently models inter-variable relationships, addressing the limitations of attention mechanisms in handling weak or irrelevant correlations; and (3) We integrate these techniques into TimesMR, advancing time series model design by highlighting that simplicity and efficiency can coexist with strong predictive performance.
> >
> > We hope this clarifies the main research contributions of our work. We look forward to your further feedback.

---

### Official Review · Reviewer_2re2 · 2025-11-01

**Soundness:** 2
**Presentation:** 2
**Contribution:** 2
**Rating:** 6
**Confidence:** 3

**Summary:**

The paper proposes TimesMR, a forecasting model for multivariate time series. It combines: 1) Multi-scale MLP modules (multi-patch and multi-downsampling) to capture temporal patterns efficiently, and 2) Grouped BiRNNs to model relationships between variables while keeping computation manageable. Across extensive experiments, TimesMR achieves strong performance and scales better than attention-based models. Its components can also plug into other backbones and improve them.

**Strengths:**

1. The proposed Grouped BiRNN offers an effective and efficient way to model relationships among variables, reducing the computational burden of attention-based methods while preserving performance.

2. The multi-scale MLP modules effectively capture temporal patterns with lightweight operations, demonstrating that complex architectures are not always necessary for strong results.

3. Extensive experiments on 16 datasets, thorough ablations, and plug-and-play validations confirm both the effectiveness and generality of the proposed components.

4. The paper is clearly written, well-organized, and convincingly shows that TimesMR achieves a strong balance between accuracy, efficiency, and scalability for multivariate time series forecasting.

**Weaknesses:**

1. The MLP-based multi-scale part feels like a lighter version of existing local time series embedding methods. The novelty mainly lies in simplifying.

2. The grouping method is more like heuristic. Variables are grouped by simple rules (√C or fixed G). There seems no discussion of how group splitting or variable order affects results.

3. The reason for using a BiRNN instead of other sequence-mixing methods in the Variable Correlation Module is not very clear to me. It seems that using attention or MLP layers could achieve similar effects: since the mixing occurs along the variable dimension, attention would not suffer from the quadratic complexity issue that arises from growing sequence length. Moreover, because the number of variables is fixed, an MLP could also perform this mixing effectively. More discussion is needed to justify this design choice and to better support the paper’s claim of “unlocking the potential of RNNs” in the title.

4. Although many recent works tend to use a short input length of 96 for simplicity, in my opinion, conducting experiments with longer input lengths would provide more practical and informative comparisons (for example, the default input length of 512 used in PatchTST). Since other sequence modeling fields, such as NLP, have already explored training with extremely long context windows, time series forecasting, as an area that often deals with relatively small datasets, could also benefit from revisiting such experimental settings. Moreover, because the searched lookback length is one of the key motivations of the proposed method, I believe it would likely perform even better with longer input lengths.

**Questions:**

1. Will the grouping method affect the final result significantly? Like using random, contiguous, or correlation-based grouping?

2. Why use an element-wise product for combining intra-group and inter-group features? It seems that additive would be more natural since the module can be seen as performing feature preprocessing?

3. The sturcture of the predictor can be more detailed. A practical guidance on choosing the predictor, the patch and down variants would be helpful.

4. Notation: It seems that the RHS in Eq. 5 appears twice?

---

> ### Author Response · Authors · 2025-11-25
>
> We thank the reviewer for the constructive feedback and we have addressed all the comments below.
>
> #### **W1. The MLP-based multi-scale part feels like a lighter version of existing local time series embedding methods. The novelty mainly lies in simplifying.**
>
> **Response:** We would like to clarify that the primary goal of our work is to design a simple yet effective architecture for multivariate time series forecasting, in contrast to existing complex models. Achieving simplicity is often more challenging than complexity, as it requires careful consideration to retain effectiveness while minimizing unnecessary components.
>
> Specifically, our MLP-based multi-scale module in Section 4.1 is designed to efficiently capture effective temporal features. While prior local time series embedding methods have demonstrated strong performance, they typically rely on heavy components such as attention mechanisms, which incur high computational costs. In contrast, our proposed multi-scale MLP module, when combined with multi-scale techniques, shows that a lightweight MLP is already capable of capturing rich temporal patterns. This is empirically supported by observations in Figure 1, where our approach achieves competitive or superior performance with significantly reduced time and memory requirements (as shown in **Table 11 and Table 12 in the revised paper**).
>
> Furthermore, as demonstrated in Figure 3 of the revised paper, the architecture of TimesMR is carefully designed to balance simplicity and effectiveness by integrating the MLP-based multi-scale module with the proposed variable correlation module. The variable correlation module in Section 4.2 introduces a novel grouped bidirectional RNN (Grouped BiRNN) approach for modeling variable relationships. Bidirectional RNNs aggregate relevant information from other variables and suppress noise from weakly correlated ones. For high-dimensional data, variables are grouped, and intra-group correlations are captured by applying bidirectional RNNs within each group. Inter-group information is exchanged via the final hidden states of each group's RNN, significantly reducing memory and computation costs while preserving effective correlation modeling (as shown in **Figure 11**).
>
> Overall, the results in Section 5.2 demonstrate the superior performance of our TimesMR model compared to other baselines, including the multi-scale MLP module and the variable correlation module. In Section 5.3 and the Appendix, we provide extensive experimental analysis to demonstrate the efficiency of the proposed method and validate the effectiveness of each component in our model.

---

> > ### Author Response · Authors · 2025-11-25
> >
> > #### **W2&Q1. How group splitting or variable order affects results. Like using random, contiguous, or correlation-based grouping?**
> >
> > **Response:** We clarify that we do not explicitly utilize any prior ordering or domain knowledge of the variables when forming groups; instead, we follow the input order of the variables as presented in the dataset. Thus, variables are grouped sequentially according to their original dataset order. In multivariate time series forecasting, variable order can be arbitrary in real-world scenarios. Our approach is intentionally designed to avoid dependence on any specific variable ordering or prior knowledge of variable relationships, ensuring general applicability and robustness.
> >
> > Specifically, suppose there are $C$ variables and we want to divide them into $G$ groups with $g$ variables per group. We first pad the variable set from $C$ to $C'$, so that $C' = G \times g$. Then, the first $g$ variables in $C'$ form the first group, the next $g$ variables form the second group, and so on until all groups are assigned.
> >
> > To further assess whether the grouping strategy affects performance, we additionally consider two alternative grouping methods: (1) Random grouping, where the variable order is randomly permuted before grouping, resulting in each group containing an arbitrary set of variables. (2) Correlation-based grouping, where variables are clustered according to their pairwise correlations, with the number of clusters set equal to the number of groups, so highly correlated variables are placed within the same group.
> >
> > To provide a comprehensive evaluation, we include eight different datasets and report the results in **Table 1 below**. We observe that both the random and correlation-based grouping strategies result in only minimal changes to our model's performance. This demonstrates that our method robustly captures dependencies among variables and is not sensitive to the specific grouping configuration, indicating its ability to model inter-variable relationships without relying on any particular grouping scheme.
> >
> > **Table 1. Comparison of the different grouping methods**
> > |Model|metric|Traffic|ECL|PEMS-BAY|METR-LA|PEMS03|PEMS04|PEMS07|PEMS08
> > |---|---|---|---|---|---|---|---|---|---|
> > |TimesMR/patch|MSE|0.426|0.161|0.365|0.735|0.097|0.084|0.073|0.108|
> > ||MAE|0.280|0.255|0.273|0.460|0.197|0.186|0.171|0.198|
> > |random|MSE|0.428|0.160|0.364|0.732|0.092|0.083|0.072|0.108|
> > ||MAE|0.282|0.255|0.275|0.470|0.196|0.186|0.169|0.201|
> > |correlation-based|MSE|0.431|0.164|0.368|0.734|0.094|0.083|0.072|0.112|
> > ||MAE|0.280|0.259|0.274|0.458|0.198|0.185|0.167|0.199|

---

> > > ### Author Response · Authors · 2025-11-25
> > >
> > > #### **W3. The reason for using a BiRNN instead of other sequence-mixing methods in the Variable Correlation Module is not very clear to me. It seems that using attention or MLP layers could achieve similar effects: since the mixing occurs along the variable dimension, attention would not suffer from the quadratic complexity issue that arises from growing sequence length. Moreover, because the number of variables is fixed, an MLP could also perform this mixing effectively. More discussion is needed to justify this design choice and to better support the paper’s claim of "unlocking the potential of RNNs" in the title.**
> > >
> > >
> > > **Response:** As suggested, we provide a detailed discussion on the choice of using BiRNNs for modeling variable correlations in Section 4.2, compared to attention mechanisms and MLP layers. In principle, both attention mechanisms and MLP layers can be used to model correlations across variables. However, each of these alternatives comes with important limitations in multivariate time-series modeling.
> > >
> > > For attention, as shown in **Figure 2**, although attention is expressive, multivariate time series often exhibit highly complex inter-variable relationships. When strong correlations exist, attention works well. However, when correlations are weak or even unrelated, which is common in datasets like ETTm1/ETTm2, attention tends to degrade performance because it inevitably incorporates large amounts of weakly correlated or irrelevant variable information. In Figure 2, removing the attention-based mixing actually improved performance on several benchmarks. Moreover, attention suffers from quadratic complexity with respect to the number of variables. While this is not an issue along the sequence length, it becomes prohibitive when the variable number is large (e.g., 862 variables in Traffic dataset, 883 in PEMS07 dataset). As shown in **Figure 11**, attention-based variable mixing incurs substantially higher time and memory cost compared to our approach.
> > >
> > > For MLP, MLP layers are lightweight, but their representational capacity is limited when dealing with complex variable correlation relationships. Our comparisons in **Figure 11** show that pure MLP mixing consistently underperforms, especially when variables exhibit a mixture of strong, weak, and even unrelated correlations. MLP layers do not have the mechanism to selectively emphasize informative variables while suppressing noisy ones.
> > >
> > > **Why grouped BiRNN?** We adopt RNNs for modeling variable relationships and propose the grouped BiRNN technique because, compared to attention and MLPs, RNNs are better suited to capture the complex dependencies present in multivariate time series. In practice, some variables exhibit strong correlations while others may be weakly correlated or even unrelated (see **Appendix A**). Achieving strong performance requires a mechanism that can emphasize informative correlations while suppressing the influence of weak or irrelevant variables. RNN architectures such as LSTMs and GRUs are particularly effective in this regard—their gating mechanisms filter out unrelated interference and selectively retain relevant information, making them well-suited for modeling intricate variable relationships. For this reason, we explore the underutilized potential of RNNs for variable correlation modeling as an alternative to attention and MLPs. Moreover, we design the Grouped BiRNN to optimize BiRNN processing along the variable dimension and accelerate training. By grouping variables, we enable parallel BiRNN operations within each group to capture intra-group interactions. The BiRNN outputs represent intra-group information exchange, while the last hidden states aggregate group-level correlations, facilitating inter-group information passing. Unlike attention, which incurs quadratic complexity when modeling relationships among variables, our grouped BiRNN method has a computational complexity of $\mathcal{O}(\sqrt{C} \cdot D^{2})$ (see **Appendix D**), scaling with the square root of the number of variables $C$.

---

> > > > ### Author Response · Authors · 2025-11-25
> > > >
> > > > #### **W4. Although many recent works tend to use a short input length of 96 for simplicity, in my opinion, conducting experiments with longer input lengths would provide more practical and informative comparisons. Because the searched lookback length is one of the key motivations of the proposed method, I believe it would likely perform even better with longer input lengths.**
> > > >
> > > > **Response:** We address this comment by conducting the new experiments.
> > > >
> > > > First, in **Figure 4 of the revised paper (Section 5.3)**, we report results for varying input lengths in {96, 192, 336, 576, 720}, thus including longer input lengths beyond 96. **Figure 4** demonstrates that as the input length increases, our method consistently achieves improved MSE, higher efficiency, and reduced training time and memory usage.
> > > >
> > > > Second, we further compare our model with baselines under the searched lookback length setting, following protocols established in prior works such as PatchTST, TimeMixer, and CycleNet. In this setting, the lookback length is selected from {96, 192, 336, 576, 720} to allow for longer inputs, and the best-performing length is reported for each method. To ensure fairness, we adopt the same protocol for our model and use the published results for the baselines. The results are summarized in **Table 2 below**. Even under the searched lookback length setting, our model TimesMR achieves the best performance in most cases.
> > > >
> > > > The results in **Figure 4 of the Section 5.3** and **Table 2 below** consistently demonstrate the effectiveness of our method for handling longer input lengths.
> > > >
> > > >
> > > > **Table 2. Performance comparison under searched lookback length**
> > > > |Datasets|metric|TimesMR/patch|ModernTCN|CycleNet|SparseTSF|TimeMixer|PatchTST|DLinear|
> > > > |---|---|---|---|---|---|---|---|---|
> > > > |ETTm1|MSE|0.344|0.351|0.362|0.361|0.348|0.353|0.357||
> > > > ||MAE|0.379|0.381|0.389|0.382|0.375|0.382|0.379||
> > > > |ETTm2|MSE|0.250|0.253|0.263|0.251|0.256|0.256|0.267|
> > > > ||MAE|0.311|0.314|0.324|0.312|0.315|0.317|0.332||||
> > > > |ETTh1|MSE|0.403|0.404|0.439|0.406|0.411|0.413|0.423|||
> > > > ||MAE|0.424|0.420|0.450|0.419|0.423|0.434|0.437||||
> > > > |ETTh2|MSE|0.336|0.322|0.367|0.344|0.316|0.324|0.431|||
> > > > ||MAE|0.384|0.379|0.401|0.386|0.384|0.381|0.447||||
> > > > |Weather|MSE|0.219|0.224|0.224|0.240|0.222|0.241|0.246|||
> > > > ||MAE|0.258|0.264|0.266|0.280|0.262|0.264|0.300||||
> > > > |Solar|MSE|0.185|0.228|0.189|0.220|0.192|0.256|0.329|||
> > > > ||MAE|0.240|0.282|0.247|0.256|0.244|0.298|0.400||||
> > > > |Traffic|MSE|0.364|0.396|0.403|0.412|0.387|0.391|0.434|||
> > > > ||MAE|0.259|0.270|0.282|0.278|0.262|0.264|0.295||||
> > > > |ECL|MSE|0.152|0.156|0.157|0.165|0.156|0.159|0.166|||
> > > > ||MAE|0.248|0.253|0.252|0.258|0.246|0.253|0.264||||
> > > > |$1^{st}$ count||**11**|1|0|1|3|0|0|
> > > >
> > > >
> > > > #### **Q2. Why use an element-wise product for combining intra-group and inter-group features? It seems that additive would be more natural since the module can be seen as performing feature preprocessing?**
> > > >
> > > >
> > > > **Response:** We adopt the element-wise product as the default mechanism for combining intra-group and inter-group features, but consider the choice of other combination method to be flexible. To assess this, we also compare the performance of the additive strategy.
> > > >
> > > > As summarized in **Table 3 below**, both the element-wise product in our method TimesMR and the additive strategies achieve similar performance across most datasets. The element-wise product method slightly outperforms the additive method on the first seven datasets, while the additive method performs marginally better on the PEMS07 and PEMS08 datasets. Overall, the performance differences between the two methods are minimal.
> > > >
> > > > This indicates that either combination strategy is suitable for our approach, and the effectiveness of our method primarily stems from the MLP-based multi-scale technique and the Grouped BiRNN module.
> > > >
> > > > **Table 3. Performance comparison with additive method**
> > > > |Model|metric|Traffic|ECL|PEMS-BAY|METR-LA|PEMS03|PEMS04|PEMS07|PEMS08
> > > > |---|---|---|---|---|---|---|---|---|---|
> > > > |TimesMR/patch|MSE|0.426|0.161|0.365|0.735|0.097|0.084|0.073|0.108|
> > > > ||MAE|0.280|0.255|0.273|0.460|0.197|0.186|0.171|0.198|
> > > > |additive|MSE|0.428|0.163|0.370|0.759|0.097|0.085|0.072|0.105|
> > > > ||MAE|0.282|0.258|0.276|0.480|0.199|0.189|0.171|0.196|

---

> > > > > ### Author Response · Authors · 2025-11-25
> > > > >
> > > > > #### **Q3. The sturcture of the predictor can be more detailed. A practical guidance on choosing the predictor, the patch and downsampling variants would be helpful.**
> > > > > **Response:** As suggested, we further elaborate on the predictor and provide practical guidance on choosing between the patch and downsamping variants in our Multi-scale Module.
> > > > >
> > > > > The predictor in our model applies a linear transformation along the hidden dimension to the final variable representations $\mathcal{X_var} \in \mathbb{R}^{C \times D}$, producing the final forecast $\hat{\mathcal{X}}_{t+1:t+H} \in \mathbb{R}^{C \times H}$, where $C$ is the number of variables, $D$ is the hidden dimension, and $H$ is the forecasting horizon.
> > > > >
> > > > > For practical guidance, as analyzed in **Appendix C.1**, we provide a clear comparison of the patch and downsamping variants in our Multi-scale Module. Specifically, we plot fragments of datasets with different characteristics and report the performance of both variants in **Figure 9**. The results show that the patch variant effectively captures local temporal variations by segmenting the input sequence into non-overlapping patches and extracting intra-patch features via MLPs. This enables the model to focus on fine-grained, high-frequency patterns such as local fluctuations or short-term changes (e.g., ETTm1 and Traffic datasets). In contrast, the downsamping variant targets global or long-term temporal patterns by reducing temporal resolution through downsampling, which aggregates information over larger time spans and emphasizes overall trends and long-term changes (e.g., ETTh2 and METR-LA datasets).
> > > > >
> > > > >
> > > > > #### **Q4. Notation: It seems that the RHS in Eq. 5 appears twice?**
> > > > > **Response:** Thank you for pointing this out. We have fixed this in the revised paper.

---

### Author Response · Authors · 2025-12-02
**Rebuttal summary**

We have addressed all reviewer comments with extensive experiments and clarifications. We thank the reviewers for their insightful comments, and the ACs, SACs, and PCs for their time and effort.

Multivariate time series forecasting is critical across various domains. Existing multi-scale models, while effective for modeling temporal dependencies, often rely on complex and computationally expensive feature extractors. Similarly, attention mechanisms, though powerful for capturing variable relationships, suffer from high computational complexity on high-dimensional datasets and can introduce noise from weakly related variables, leading to performance degradation.

To address these challenges, we propose a novel method, TimesMR. Our contributions, as recognized by the reviewers, are summarized as follows:

1. **Effective and efficient multi-scale MLP modules.** The multi-scale MLP modules capture temporal patterns using lightweight operations, replacing complex temporal feature extractors with two efficient variants—Multi-patch MLP and Multi-downsampling MLP—to capture multi-scale temporal dynamics **(Reviewer 2re2, vvAx)**.

2. **Effective and efficient Grouped BiRNN.** The proposed Grouped BiRNN provides an efficient way to model relationships among variables, significantly reducing the computational burden of attention-based methods to $\mathcal{O}(\sqrt{C})$ while preserving performance. It demonstrates a compelling trade-off between accuracy and efficiency, striking a good balance in high-dimensional settings where attention can be costly or noisy, and achieves substantially faster training times on large datasets **(Reviewer 2re2, H2Md, vvAx)**.

3. **Strong empirical performance.** TimesMR consistently outperforms 18 strong baselines, including iTransformer, Pathformer, and TimeMixer, across 16 datasets, providing solid and convincing empirical evidence for the proposed model **(Reviewer 2re2, H2Md, vvAx)**.

4. **Comprehensive component-wise analysis.** The paper validates the plug-and-play capability of its modules and provides a direct comparison of Grouped BiRNN against Grouped Attention, along with a cost-benefit analysis for different group numbers, all of which strongly support the paper's claims. These validations further confirm the effectiveness and generality of the proposed components **(Reviewer 2re2, H2Md, vvAx)**.

5. **Clearly written and well-organized architecture.** The paper is clearly written and well organized, convincingly showing that TimesMR achieves a strong balance between accuracy, efficiency, and scalability for multivariate time series forecasting. The architecture and data flow—including patching and downsampling, RevIN, Grouped BiRNN intra- and inter-passes, and fusion—are clearly illustrated **(Reviewer 2re2, H2Md)**.

In the rebuttal, we have addressed the reviewers' comments, summarized below:

1. **For Reviewer 2re2:** We conducted experiments on grouping methods (W2, Q1), compared performance under different searched lookback lengths (W4), applied an element-wise product to combine intra-group and inter-group features (Q2), and provided detailed analyses of the MLP-based multi-scale component (W1), the rationale for using a BiRNN (W3), guidance on choosing the predictor, and the patch and downsampling variants (Q3). Although we have not received further responses, we believe all concerns have been adequately addressed.

2. **For Reviewer H2Md:** We conducted experiments with stronger diagnostics, including partial correlation and spectral coherence (W2), analyzed per-dataset standard deviations (Q2), performed paired significance tests (Q2), and provided detailed explanations regarding the paper's contributions (W1), non-attention baselines, foundation-model-style forecasters (W3), lookback length, and experimental protocol (Q1). **These efforts effectively addressed the reviewer's concerns regarding the technical aspects.** Regarding Reviewer H2Md's further concern about the main research claims beyond empirical superiority, we have provided detailed explanations and believe this concern has been adequately addressed.

3. **For Reviewer vvAx:** We conducted experiments on the grouping strategy (W2, Q2) and different padding strategies (W3, Q3), and provided detailed explanations regarding architectural redundancy (W1, Q1) and the motivation for BiRNN integration (W4). Although we have not received further responses, we believe all concerns have been adequately addressed.

We thank all reviewers, ACs, SACs, and PCs for their time and efforts.

Best,

All authors

---

### Meta-Review · Area_Chair_jyqS · 2026-01-07

**Summary:**

The authors propose a forecasting model for multivariate time series. It introduces two modules, i.e., Multi-scale MLP modules (multi-patch MLP and multi-downsampling MLP) to capture temporal patterns, and the Grouped BiRNNs to model relationships among variables, reducing the computational burden of attention-based methods while preserving performance. Extensive and comprehensive experiments demonstrate that the TimesMR achieves strong performance and scales better than attention-based models.


Reviewers acknowledged the empirical performance of the proposed method. The main concerns include the technical novelty of the method, the design of the grouping method, the choice of BiRNN, longer input lengths, and evaluation metric selection etc. During the rebuttal, the authors substantially strengthened their paper by addressing key methodological concerns with new experiments and clarifications: (i) more grouping methods for comparison with random and correlation-based grouping in details to demonstrate the low sensitivity; (ii) padding/truncation strategy disclosure and robustness across padding variants; (iii) longer-input and searched-lookback protocols with clearer training/validation procedures to mitigate leakage concerns; and (iv) stronger correlation/noise diagnostics (e.g., partial correlation and spectral coherence) and stability/significance analyses over multiple seeds. These additions improve credibility and reproducibility.

The remaining weakness is that the main research claim, i.e., “unlocking the potential of RNNs” lacks a detailed and comprehensive theoretical or mechanistic characterization. Instead, the authors only demonstrate accuracy-efficiency trade-offs. Therefore, this paper is primarily empirical/engineering-oriented, and the justification for the architectural design remains somewhat empirical. Given the strong experimental evidence, clear presentation, and the thorough rebuttal addressing most technical risks, I recommend rejection at this time, with encouragement to substantially revise and resubmit to a future venue.

**Reviewer Concerns:**

The rebuttal adequately addressed most of the reviewers’ technical and methodological concerns, including: (i) clarification and empirical validation of the grouping strategy (sequential vs. random vs. correlation-based), showing low sensitivity; (ii) explicit description and analysis of the padding–truncate mechanism, with experiments demonstrating negligible impact; (iii) justification of the Grouped BiRNN over attention/MLP via efficiency, robustness to weak correlations, and added ablations; (iv) experiments with longer and searched lookback lengths, clear validation splits, and no test leakage; and (v) added stability and significance analyses plus stronger correlation diagnostics.

The main outstanding concern is conceptual rather than technical. Specifically, despite improved explanations, the paper’s core research claim remains primarily empirical/engineering-focused, and the architectural motivation (especially the broader claim of “unlocking the potential of RNNs”) is still qualitative, lacking a detailed and comprehensive theoretical or mechanistic characterization beyond demonstrated accuracy-efficiency trade-offs.

**Reviewer Scores:**

+ Reviewer 2re2 would likely have maintained or slightly increased his/her score, as all of their specific weaknesses and questions, i.e., grouping strategy, BiRNN motivation, lookback length, and architectural details, were directly addressed with additional experiments and clarifications.
+ Reviewer H2Md would likely have maintained his/her score, since their technical concerns about diagnostics, variance, significance testing, and evaluation protocol were convincingly resolved, though they may still view the contribution as primarily empirical.
+ Reviewer vvAx would likely have maintained his/her score modestly, as the rebuttal explicitly clarified grouping, padding, redundancy between multi-scale modules, and the efficiency-accuracy trade-off, even if they might still perceive the design as an engineering optimization rather than a fundamentally new modeling paradigm.

---

### Decision · Program_Chairs · 2026-01-26

Reject